# Inhibitory proteins block substrate access by occupying the active site cleft of *Bacillus subtilis* intramembrane protease SpoIVFB

**Sandra Olenic, Lim Heo, Michael Feig, Lee Kroos\***

Michigan State University, East Lansing, United States

**Abstract** Intramembrane proteases (IPs) function in numerous signaling pathways that impact health, but elucidating the regulation of membrane-embedded proteases is challenging. We examined inhibition of intramembrane metalloprotease SpoIVFB by proteins BofA and SpoIVFA. We found that SpoIVFB inhibition requires BofA residues in and near a predicted transmembrane segment (TMS). This segment of BofA occupies the SpoIVFB active site cleft based on cross-linking experiments. SpoIVFB inhibition also requires SpoIVFA. The inhibitory proteins block access of the substrate N-terminal region to the membrane-embedded SpoIVFB active site, based on additional cross-linking experiments; however, the inhibitory proteins did not prevent interaction between the substrate C-terminal region and the SpoIVFB soluble domain. We built a structural model of SpoIVFB in complex with BofA and parts of SpoIVFA and substrate, using partial homology and constraints from cross-linking and co-evolutionary analyses. The model predicts that conserved BofA residues interact to stabilize a TMS and a membrane-embedded C-terminal region. The model also predicts that SpoIVFA bridges the BofA C-terminal region and SpoIVFB, forming a membrane-embedded inhibition complex. Our results reveal a novel mechanism of IP inhibition with clear implications for relief from inhibition in vivo and design of inhibitors as potential therapeutics.

**\*For correspondence:**
kroos@msu.edu

**Competing interest:** The authors declare that no competing interests exist.

## Editor's evaluation

A member of a family of metalloproteases conserved in all three domains of life, SpoIVFB is required for development in the spore-forming firmicute, *Bacillus subtilis*. SpoIVFB activity is tightly controlled, however, its status as a multipass membrane protein has made illuminating the molecular basis of SpoIVFB inhibition challenging. Here, Olenic and colleagues combine genetics, cross-linking, and co-evolutionary analysis to develop a structural model of interaction between SpoIVFB and its inhibitors SpoIVFA and BofA. Given the conservation and importance of this family of metalloproteases, this work should have a broad impact influencing our understanding of the regulation of this class of proteins across the tree of life.

## Introduction

Intramembrane proteases (IPs) are membrane proteins containing a membrane-embedded active site. IPs cleave membrane-associated substrates within a transmembrane segment (TMS) or near the membrane surface during the process of regulated intramembrane proteolysis (RIP) (*Brown et al., 2000*). Released substrate fragments impact diverse signaling pathways in a wide variety of organisms (*Urban, 2013*). IPs also profoundly affect protein degradation (*Kühnle et al., 2019*). The four known IP families are metallo IPs like SpoIVFB, aspartyl IPs like γ-secretase, serine IPs (rhomboids),

**eLife digest** Proteases are a type of protein that work by cutting up other proteins. The part of the protease that does the cutting is called the active site. Intramembrane proteases are a specific group of proteases that cut up the proteins within cell membranes. There is a lot of interest in learning how to control intramembrane proteases because they are important in regulating the signaling processes that cells use to communicate. SpoIVFB is an intramembrane protease from the bacterium *Bacillus subtilis* that is studied often as a model for these types of proteases.

*Bacillus subtilis* uses SpoIVFB to produce spores, dormant reproductive cells that can survive extreme, harsh conditions for long periods with minimal energy. SpoIVFB is part of the system that allows spores to communicate with their 'parent cells', the cells they develop in. The activity of this protein is blocked by two other proteins called SpoIVFA and BofA. When these proteins are destroyed, SpoIVFB becomes active, but it is unclear exactly how SpoIVFA and BofA inhibit SpoIVFB. Understanding this relationship could help to reveal ways to regulate other intramembrane proteases.

To address this question, Olenic et al. used genetic, biochemical and computer modelling techniques to study how SpoIVFB activity is regulated in *Bacillus subtilis.* The results show that a region of BofA blocks the area of SpoIVFB that cuts a protein called Pro-σ$^K$, which stops SpoIVFB from releasing active σ$^K$ into the 'parent cell'. By making genetic variants of BofA, Olenic et al. identified three parts of BofA that are needed to fully inhibit SpoIVFB. A computer model predicts that these three parts give BofA the right shape to inhibit SpoIVFB, and that SpoIVFA helps by forming a bridge between BofA and SpoIVFB.

This investigation reveals how the intramembrane protease SpoIVFB is regulated by SpoIVFA and BofA. This information could be useful in developing inhibitors for other intramembrane proteases. The next stage will be to make and test artificial inhibitors based on the structures studied here. If successful, these could have applications in areas such as medicine, agriculture, industry and environmental protection.

and glutamyl IP Rce1 (*Urban, 2013*; *Manolaridis et al., 2013*). Crystal structures of one or more IP in each family (*Manolaridis et al., 2013*; *Hu et al., 2011*; *Li et al., 2012*; *Feng et al., 2007*; *Wang et al., 2006*; *Wu et al., 2006*) reveal that TMSs arrange to form a channel that delivers water to the active site for hydrolysis of a substrate peptide bond. Structures of rhomboid·peptide (inhibitor and substrate) complexes (*Cho et al., 2016*; *Cho et al., 2019*; *Tichá et al., 2017*) and γ-secretase·substrate complexes (*Yang et al., 2019*; *Zhou et al., 2019*) help guide the design of IP modulators as therapeutics.

Metallo IPs activate transcription factors via RIP in all three domains of life (*Brown et al., 2000*; *Urban, 2013*). S2P regulates mammalian cholesterol homeostasis and responses to endoplasmic reticulum (ER) stress and viral infection (*Rawson, 2013*; *Ye, 2013*). S2P homologs play important roles in plant chloroplast development (*Adam, 2013*; *Chang et al., 2007*) and fungal virulence (*Adam, 2013*; *Chang et al., 2007*). Bacterial metallo IPs enhance pathogenicity, control stress responses and polar morphogenesis, produce mating signals, and clear signal peptides from the membrane (*Urban, 2009*; *Kroos and Akiyama, 2013*; *Schneider and Glickman, 2013*). SpoIVFB is a bacterial metallo IP that cleaves inactive Pro-σ$^K$ to σ$^K$, which directs RNA polymerase to transcribe genes necessary for endospore formation of *Bacillus subtilis* (*Kroos and Akiyama, 2013*). Other bacilli and most clostria likewise require SpoIVFB homologs to form endospores (*de Hoon et al., 2010*; *Galperin et al., 2012*; *Abecasis et al., 2013*; *Ramos-Silva et al., 2019*). Endospores are dormant and are able to survive harsh environmental conditions (*McKenney et al., 2013*), enhancing the persistence of pathogenic species (*Al-Hinai et al., 2015*; *Checinska et al., 2015*; *Browne et al., 2016*). Understanding the regulation of SpoIVFB could lead to new strategies to manipulate endosporulation and other processes involving IPs in bacteria and eukaryotes.

During endospore formation, *B. subtilis* forms a polar septum that divides the mother cell (MC) and forespore (FS) compartments (*Tan and Ramamurthi, 2014*; *Figure 1—figure supplement 1*). The MC engulfs the FS, surrounding it with a second membrane and pinching it off within the MC. Regulation of SpoIVFB involves inhibition by BofA and SpoIVFA (*Cutting et al., 1990*; *Cutting et al., 1991b*; *Ricca et al., 1992*). The three proteins form a complex in the outer FS membrane during engulfment

(*Resnekov et al., 1996*; *Rudner et al., 2002*; *Rudner and Losick, 2002*). BofA appears to be the direct inhibitor of SpoIVFB (*Zhou and Kroos, 2004*), while SpoIVFA appears to localize and stabilize the inhibition complex (*Rudner and Losick, 2002*; *Kroos et al., 2002*). Signaling from the FS relieves inhibition of SpoIVFB (*Figure 1—figure supplement 1*). The FS exports two proteases, SpoIVB and CtpB, into the space between the two membranes surrounding the FS (*Cutting et al., 1991a*; *Pan et al., 2003*). SpoIVB cleaves the C-terminal end of SpoIVFA (*Dong and Cutting, 2003*; *Campo and Rudner, 2006*; *Campo and Rudner, 2007*; *Mastny et al., 2013*) and CtpB can cleave the C-terminal ends of both SpoIVFA and BofA (*Campo and Rudner, 2006*; *Campo and Rudner, 2007*; *Zhou and Kroos, 2005*). Once inhibition is relieved, SpoIVFB cleaves the N-terminal 21-residue pro-sequence from Pro-$\sigma^K$, releasing $\sigma^K$ into the MC (*Rudner et al., 1999*; *Yu and Kroos, 2000*; *Zhou et al., 2009*). $\sigma^K$ directs RNA polymerase to transcribe genes whose products form the spore coat and lyse the MC, releasing a mature spore (*Kroos et al., 1989*; *Eichenberger et al., 2004*).

Regulation of SpoIVFB by direct interaction with the inhibitory proteins BofA and SpoIVFA differs from known regulation of other IPs. In eukaryotic cells, some IP substrates initially localize to a different organelle than their cognate IP, and regulation involves membrane trafficking to deliver the substrate to the enzyme (*Kühnle et al., 2019*; *Morohashi and Tomita, 2013*). In bacteria and eukaryotes, many IP substrates require an initial extramembrane cleavage by another protease prior to the intramembrane cleavage, and the initial cleavage is typically the regulated step (*Urban, 2009*; *Kroos and Akiyama, 2013*; *Schneider and Glickman, 2013*; *Lichtenthaler et al., 2018*; *Beard et al., 2019*). In contrast, SpoIVFB cleaves Pro-$\sigma^K$ only once. BofA and SpoIVFA inhibit SpoIVFB by an unknown mechanism. Understanding the mechanism of inhibition could have broad implications since the catalytic core of metallo IPs includes three conserved TMSs (*Kinch et al., 2006*) and general principles may emerge that inform potential inhibition strategies for other IP types.

Our results reveal a novel mechanism of IP inhibition in which the second predicted TMS (TMS2) of BofA occupies the SpoIVFB active site cleft. Together BofA and SpoIVFA block access of the N-terminal pro-sequence region (Proregion) of Pro-$\sigma^K$ to the membrane-embedded SpoIVFB active site, yet the C-terminal soluble regions of Pro-$\sigma^K$ and SpoIVFB can interact. We used these results in combination with prior cross-linking studies (*Zhang et al., 2013*; *Halder et al., 2017*), partial homology, and evolutionary co-variation of amino acid residues, to constrain a structural model of SpoIVFB in complex with BofA and parts of SpoIVFA and Pro-$\sigma^K$. The model predicts that conserved BofA residues interact to stabilize TMS2 and its C-terminal region, and that SpoIVFA bridges the BofA C-terminal region and SpoIVFB to form an inhibition complex. The predicted inhibition complex has implications for activation of SpoIVFB and its orthologs in other endospore formers, as well as for translational efforts to design therapeutic IP inhibitors.

## Results

### SpoIVFB inhibition requires both BofA and SpoIVFA in *Escherichia coli*

To study RIP of Pro-$\sigma^K$, we engineered *Escherichia coli* to synthesize variants of SpoIVFB and Pro-$\sigma^K$, and the inhibitory proteins BofA and SpoIVFA, in various combinations. The SpoIVFB variant contains the extra TMS cytTM (*Saribas et al., 2001*; *Figure 1A*), which improves accumulation (*Zhou et al., 2009*). The substrate variant Pro-$\sigma^K$(1–127) lacks the C-terminal half of Pro-$\sigma^K$, but SpoIVFB cleaves Pro-$\sigma^K$(1–127) accurately and the cleavage product is easily separated from the substrate by SDS-PAGE (*Prince et al., 2005*). cytTM-SpoIVFB cleaved about 80% of the Pro-$\sigma^K$(1–127) upon production from a single plasmid (*Figure 1B*, lane 1). The additional production of GFPΔ27BofA, a functional fusion protein lacking the first predicted TMS (TMS1) of BofA (*Rudner and Losick, 2002*), and SpoIVFA from a second plasmid reduced cleavage to about 30% (lane 2). Production of full-length BofA (undetectable by immunoblot due to lack of antibodies) and SpoIVFA from a second plasmid gave a similar result (lane 3).

To improve inhibition of cytTM-SpoIVFB in *E. coli*, we created single 'pET Quartet' plasmids to synthesize all four proteins, reasoning that unequal copy number of the two plasmids in some cells could result in too little of the inhibitory proteins. Indeed, pET Quartets producing either GFPΔ27BofA or BofA exhibited very little Pro-$\sigma^K$(1–127) cleavage (*Figure 1B*, lanes 4 and 5). A longer exposure of the immunoblot revealed a faint cleavage product with pET Quartet GFPΔ27BofA, but not with pET Quartet BofA, indicating that full-length BofA inhibits cleavage slightly better than GFPΔ27BofA.

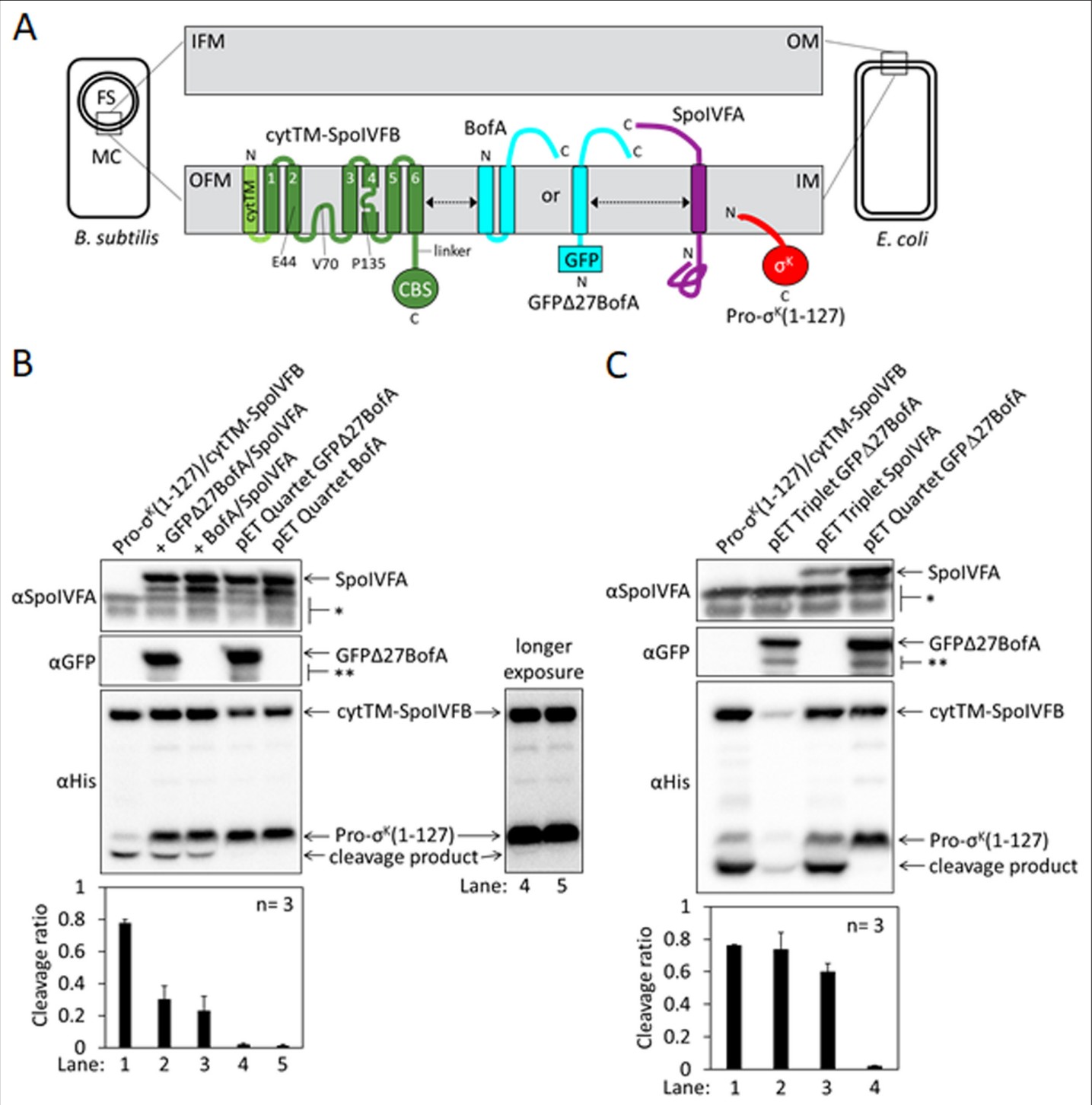

**Figure 1.** Inhibition of Pro-σ$^K$ cleavage. (**A**) Diagram of SpoIVFB inhibition during *Bacillus subtilis* endosporulation and upon heterologous expression in *Escherichia coli*. During endosporulation (*Left*), the mother cell (MC) produces SpoIVFB and its inhibitory proteins BofA and SpoIVFA, which localize to the outer forespore (FS) membrane (OFM). The MC also produces Pro-σ$^K$, which associates with membranes. When synthesized in *E. coli* (*Right*), the proteins localize to the inner membrane (IM). The expanded view of the membranes (*Center*) shows a SpoIVFB variant with an extra N-terminal transmembrane segment (cytTM), and highlights several residues (E44, V70, and P135) at or near the active site in the membrane domain, which is connected to the CBS domain by an interdomain linker. When produced in *E. coli*, cytTM-SpoIVFB cleaves Pro-σ$^K$(1–127), removing its N-terminal Proregion [Note: Pro-σ$^K$(1–126) was renamed Pro-σ$^K$(1–127) as explained in **Halder et al., 2017**]. Coproduction of SpoIVFA and either full-length BofA or GFPΔ27BofA (lacking predicted TMS1) inhibits Pro-σ$^K$(1–127) cleavage. The dashed double-headed arrows indicate that SpoIVFB, BofA or GFPΔ27BofA, and SpoIVFA form a complex of unknown structure. (**B**) Cleavage assays comparing inhibition by SpoIVFA and either GFPΔ27BofA or full-length BofA

*Figure 1 continued on next page*

*Figure 1 continued*

in *E. coli*. Pro-σ$^K$(1–127) and cytTM-SpoIVFB were produced alone (lane 1, pYZ2) or in combination with GFPΔ27BofA and SpoIVFA (lane 2, pYZ46), or full-length BofA and SpoIVFA (lane 3, pSO212). Alternatively, 'pET Quartet' plasmids were used to produce Pro-σ$^K$(1–127), cytTM-SpoIVFB, SpoIVFA, and either GFPΔ27BofA (lane 4, pSO40) or full-length BofA (lane 5, pSO213). Samples collected after 2 hr of IPTG induction were subjected to immunoblot analysis with SpoIVFA (*Top*), GFP (*Middle*), or penta-His antibodies (*Bottom*, 2 and 30 s exposures). The single star (*) indicates cross-reacting proteins below SpoIVFA and the double star (**) indicates breakdown species of GFPΔ27BofA. A breakdown species below SpoIVFA (not indicated) is observed in some experiments. The graph shows quantification of the cleavage ratio (cleavage product/[Pro-σ$^K$(1–127)+cleavage product]) for three biological replicates. Error bars, 1 standard deviation. (**C**) Cleavage assays comparing inhibition by either GFPΔ27BofA or SpoIVFA. pET Triplet plasmids were used to produce Pro-σ$^K$(1–127), cytTM-SpoIVFB, and either GFPΔ27BofA (lane 2, pSO64) or SpoIVFA (lane 3, pSO65). Samples were subjected to immunoblot analysis and quantification as in (**B**).

The online version of this article includes the following source data and figure supplement(s) for figure 1:

**Source data 1.** Immunoblot images (raw and annotated) and quantification of cleavage assays (*Figure 1B and C*).

**Figure supplement 1.** Morphological changes during endosporulation and regulated intramembrane proteolysis (RIP) of Pro-σ$^K$ in *Bacillus subtilis*.

**Figure supplement 2.** Full-length BofA alone fails to inhibit Pro-σ$^K$(1–127) cleavage in *Escherichia coli* and full-length Pro-σ$^K$ is similar to Pro-σ$^K$(1–127) in terms of requirements for cleavage inhibition.

**Figure supplement 2—source data 1.** Immunoblot images (raw and annotated) and quantification of cleavage assays.

**Figure supplement 3.** SpoIVB partially relieves inhibition of SpoIVFB by BofA and SpoIVFA in *Escherichia coli*.

**Figure supplement 3—source data 1.** Immunoblot images (raw and annotated) and quantification of cleavage assays.

**Figure supplement 4.** An F66A substitution in cytTM-SpoIVFB partially overcomes inhibition by GFPΔ27BofA and SpoIVFA in *Escherichia coli*.

**Figure supplement 4—source data 1.** Immunoblot images (raw and annotated) and quantification of cleavage assays.

To determine whether cleavage inhibition requires both inhibitory proteins, we created 'pET Triplet' plasmids to synthesize Pro-σ$^K$(1–127), cytTM-SpoIVFB, and either GFPΔ27BofA, full-length BofA, or SpoIVFA. pET Triplets containing GFPΔ27BofA (*Figure 1C*, lane 2) or BofA (*Figure 1—figure supplement 2A*, lane 2) exhibited cleavage, so cytTM-SpoIVFB inhibition requires SpoIVFA. However, less cytTM-SpoIVFB, Pro-σ$^K$(1–127), and cleavage product accumulated, suggesting that BofA inhibits synthesis and/or enhances degradation of the other proteins when SpoIVFA is absent. pET Triplet SpoIVFA also exhibited cleavage (*Figure 1C*, lane 3), so cytTM-SpoIVFB inhibition requires either GFPΔ27BofA or full-length BofA. In this case, cytTM-SpoIVFB, Pro-σ$^K$(1–127), and cleavage product accumulated normally, but less SpoIVFA accumulated, suggesting that SpoIVFA synthesis decreases and/or degradation increases when BofA is absent. We conclude that inhibition of Pro-σ$^K$(1–127) cleavage by cytTM-SpoIVFB requires both inhibitory proteins in *E. coli*. Inhibition of full-length Pro-σ$^K$ cleavage by cytTM-SpoIVFB also requires both inhibitory proteins in *E. coli* (*Figure 1—figure supplement 2B*). The heterologous system mimics the endogenous pathway of *B. subtilis* since inhibition of Pro-σ$^K$ cleavage by SpoIVFB requires both inhibitory proteins during sporulation (*Cutting et al., 1990*; *Rudner and Losick, 2002*; *Ramirez-Guadiana et al., 2018*).

The *E. coli* system also mimics the endogenous pathway with respect to the effects of SpoIVB and an F66A substitution in SpoIVFB on cleavage inhibition. SpoIVB relieves inhibition of SpoIVFB during *B. subtilis* sporulation, although efficient relief also requires CtpB (*Pan et al., 2003*; *Campo and Rudner, 2006*; *Campo and Rudner, 2007*; *Zhou and Kroos, 2005*). Addition of SpoIVB to the *E. coli* system partially relieved cytTM-SpoIVFB inhibition, resulting in 14% and 5% more cleavage in systems with GFPΔ27BofA and full-length BofA, respectively, as compared with the addition of catalytically inactive SpoIVB S378A (*Figure 1—figure supplement 3*). In the absence of SpoIVB, an F66A substitution in SpoIVFB-YFP allowed Pro-σ$^K$ cleavage during *B. subtilis* sporulation, albeit with reduced efficiency (*Ramirez-Guadiana et al., 2018*). An F66A substitution in cytTM-SpoIVFB partially relieved inhibition, resulting in 12% more cleavage in the *E. coli* system with GFPΔ27BofA (*Figure 1—figure supplement 4*). We conclude that the *E. coli* system mimics the endogenous pathway of *B. subtilis*. Since most endospore formers encode BofA, but about half lack a recognizable gene for SpoIVFA (*de Hoon et al., 2010*; *Galperin et al., 2012*), we focused on using the strong cleavage inhibition observed with pET Quartet plasmids in *E. coli* (*Figure 1B*) to identify residues of BofA important for inhibition of SpoIVFB.

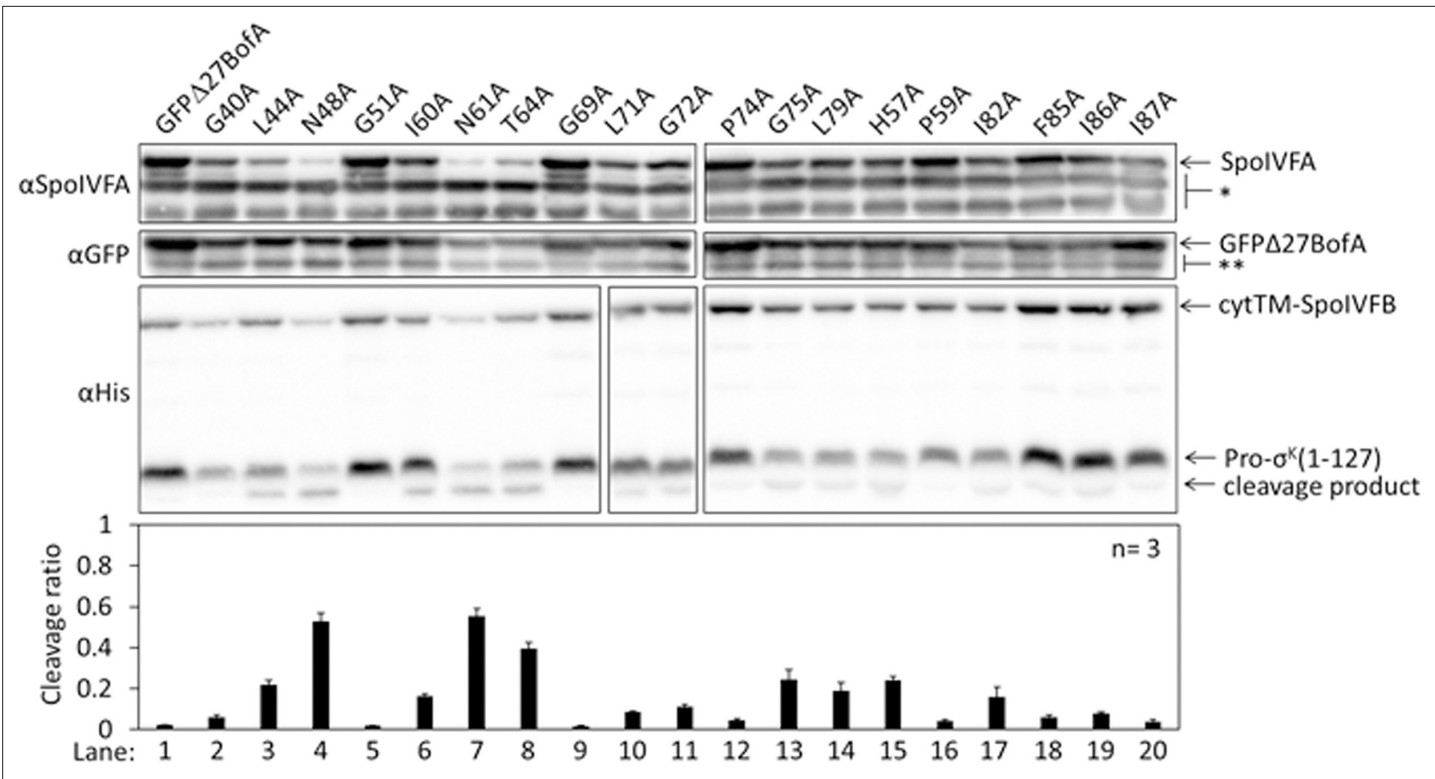

**Figure 2.** Effects of alanine substitutions in GFPΔ27BofA on inhibition of Pro-σ^K(1–127) cleavage in *Escherichia coli*. pET Quartet plasmids were used to produce Pro-σ^K(1–127), cytTM-SpoIVFB, SpoIVFA, and GFPΔ27BofA (lane 1, pSO40) or Ala-substituted GFPΔ27BofA (lanes 2–20, pSO44-pSO58 and pSO60-pSO63). Samples collected after 2 hr of IPTG induction were subjected to immunoblot analysis with SpoIVFA, GFP, and penta-His antibodies. Single (*) and double (**) stars are explained in the *Figure 1B* legend, as is the graph.

The online version of this article includes the following source data and figure supplement(s) for figure 2:

**Source data 1.** Immunoblot images (raw and annotated) and quantification of cleavage assays.

**Figure supplement 1.** Sequence alignments of BofA orthologs to determine conserved residues.

**Figure supplement 2.** GFPΔ27BofA variants accumulate normally in *Escherichia coli* in the absence of the other *Bacillus subtilis* proteins.

**Figure supplement 2—source data 1.** Immunoblot images (raw and annotated) and quantification of protein levels.

**Figure supplement 3.** BofA C-terminal residues and residues preceding predicted TMS2 contribute to inhibition of SpoIVFB in *Escherichia coli*.

**Figure supplement 3—source data 1.** Immunoblot images (raw and annotated) and quantification of cleavage assays.

## Three conserved residues of BofA are important for inhibition of SpoIVFB in *E. coli*

To identify residues of BofA that may play a role in SpoIVFB inhibition, we analyzed an alignment of 70 BofA orthologs (*Figure 2—figure supplement 1A*). Thirteen highly conserved residues reside in predicted TMS2 and the C-terminal region. Predicted TMS1 lacks highly conserved residues and GFPΔ27BofA (with a deletion of predicted TMS1) is a functional inhibitor (*Rudner and Losick, 2002*; *Zhou and Kroos, 2004*) albeit slightly less inhibitory than full-length BofA (*Figure 1B*, longer exposure). These observations suggest that TMS1 plays a minor role in SpoIVFB inhibition and that any residues compatible with TMS formation may suffice for that role. An alignment of the 31 BofA orthologs from species with a recognizable gene for SpoIVFA revealed an additional four conserved residues that we deemed of interest for Ala substitutions (*Figure 2—figure supplement 1B*). We also substituted Ala for F85 and I87 (as well as for I86, which is conserved), since deletion of three residues from the C-terminal end of BofA caused a loss of function or stability in *B. subtilis* (*Ricca et al., 1992*; *Varcamonti et al., 1997*). We made the Ala substitutions in GFPΔ27BofA and coproduced the variants with Pro-σ^K(1–127), cytTM-SpoIVFB, and SpoIVFA from pET Quartet plasmids in *E. coli*.

Three GFPΔ27BofA variants, N48A, N61A, and T64A allowed 40%–60% cleavage of Pro-σ^K(1–127), indicating that these substitutions strongly impaired inhibition of cytTM-SpoIVFB (*Figure 2*, lanes 4, 7,

and 8). We observed less of the N61A and T64A variants than most of the other GFPΔ27BofA variants. This effect depended on coproduction with the other *B. subtilis* proteins, since the three GFPΔ27BofA variants accumulated normally in *E. coli* in the absence of the other proteins (*Figure 2—figure supplement 2*). For all three variants, we observed less SpoIVFA and cytTM-SpoIVFB, perhaps indicative of altered complex formation leading to protein instability, as inferred from studies in *B. subtilis* (*Rudner and Losick, 2002*; *Resnekov, 1999*).

The other GFPΔ27BofA variants had less effect on Pro-σ$^K$(1–127) cleavage, indicating less effect on cytTM-SpoIVFB inhibition in *E. coli* (*Figure 2*). Although the F85A, I86A, and I87A variants had little effect, both a triple-Ala variant and a variant lacking residues 85–87 strongly impaired inhibition of cytTM-SpoIVFB (*Figure 2—figure supplement 3A*); the latter mimicking the results in *B. subtilis* (*Ricca et al., 1992*; *Varcamonti et al., 1997*). We also used pET Quartet plasmids in *E. coli* to show for the first time that the nine residues preceding predicted TMS2 of GFPΔ27BofA contribute to its inhibitory function, perhaps by moving the GFP tag away from the membrane (*Figure 2—figure supplement 3B*).

## The conserved residues of BofA are important for SpoIVFB inhibition during *B. subtilis* sporulation

To test the effects of the three GFPΔ27BofA variants (N48A, N61A, and T64A) during sporulation, we produced each in a *spoIVB165 bofA::erm* double mutant in which production of GFPΔ27BofA from a non-native chromosomal locus inhibits Pro-σ$^K$ cleavage by SpoIVFB (*Rudner and Losick, 2002*; *Zhou and Kroos, 2004*). The null mutation in *spoIVB* (*spoIVB165*) prevents SpoIVFA cleavage, so a *spoIVB165* single mutant exhibits little or no RIP of Pro-σ$^K$, but the double mutant exhibits unregulated Pro-σ$^K$ cleavage because the additional null mutation in *bofA* (*bofA::erm*) relieves inhibition of SpoIVFB. We produced GFPΔ27BofA and the variants in the double mutant under control of the native *bofA* promoter from the non-native *amyE* chromosomal locus. We examined the levels of GFPΔ27BofA and the variants, and SpoIVFA, SpoIVFB, Pro-σ$^K$, and σ$^K$ (i.e., cleavage product) during sporulation, and we included wild type, the *spoIVB165* single mutant, and the *spoIVB165 bofA::erm* double mutant as controls.

The controls and the double mutant producing GFPΔ27BofA gave the expected results. We observed Pro-σ$^K$ cleavage primarily between 4 and 5 hr poststarvation (PS) in wild type, very little cleavage in the *spoIVB165* single mutant, and premature cleavage at 4 hr in the *spoIVB165 bofA::erm* double mutant (*Figure 3A*, lanes 1–6). We detected very little SpoIVFA and SpoIVFB in the double mutant, consistent with the need for BofA to stabilize these proteins (*Rudner and Losick, 2002*). We observed little cleavage in the double mutant producing GFPΔ27BofA, indicative of SpoIVFB inhibition (*Figure 3A*, lanes 7 and 8). GFPΔ27BofA allowed slightly more Pro-σ$^K$ cleavage at 5 hr (lane 8) than BofA in the *spoIVB165* single mutant (lane 4), suggesting that full-length BofA is a slightly better inhibitor of SpoIVFB during *B. subtilis* sporulation, as observed in *E. coli* (*Figure 1B*, longer exposure).

Strikingly, production of the GFPΔ27BofA N48A or N61A variant in the double mutant allowed premature cleavage of Pro-σ$^K$ at 4 hr (*Figure 3A*, lanes 9 and 11), indicating loss of SpoIVFB inhibition. The levels of the variants were similar to GFPΔ27BofA at 4 hr, but reduced at 5 hr, especially for the N61A variant (lane 12). In comparison with the strain that produced GFPΔ27BofA, the SpoIVFA level was normal in the strain that produced the N48A variant, but the SpoIVFB level was very low, and the levels of both SpoIVFA and SpoIVFB were low in the strains producing the N61A and T64 variants. The reduced levels of SpoIVFB and SpoIVFA may reflect protein instability due to altered complex formation with the GFPΔ27BofA variants, as inferred from reduced SpoIVFB and SpoIVFA levels in the absence of BofA during *B. subtilis* sporulation (*Rudner and Losick, 2002*).

Production of the T64A variant in the double mutant (*Figure 3A*, lanes 13 and 14) showed a pattern similar to wild type (lanes 1 and 2); Pro-σ$^K$ cleavage increased between 4 and 5 hr, and the levels of SpoIVFA and SpoIVFB decreased. The levels of all three proteins were slightly lower in the strain producing the T64A variant than in wild type. The level of the T64A variant was similar to the N48A and N61A variants at 4 hr, yet the T64A variant allowed less Pro-σ$^K$ cleavage (lanes 9, 11, and 13), suggesting that the T64A variant inhibits SpoIVFB, although not as well as GFPΔ27BofA (lane 7). We conclude that the three conserved residues of BofA are important for SpoIVFB inhibition during sporulation.

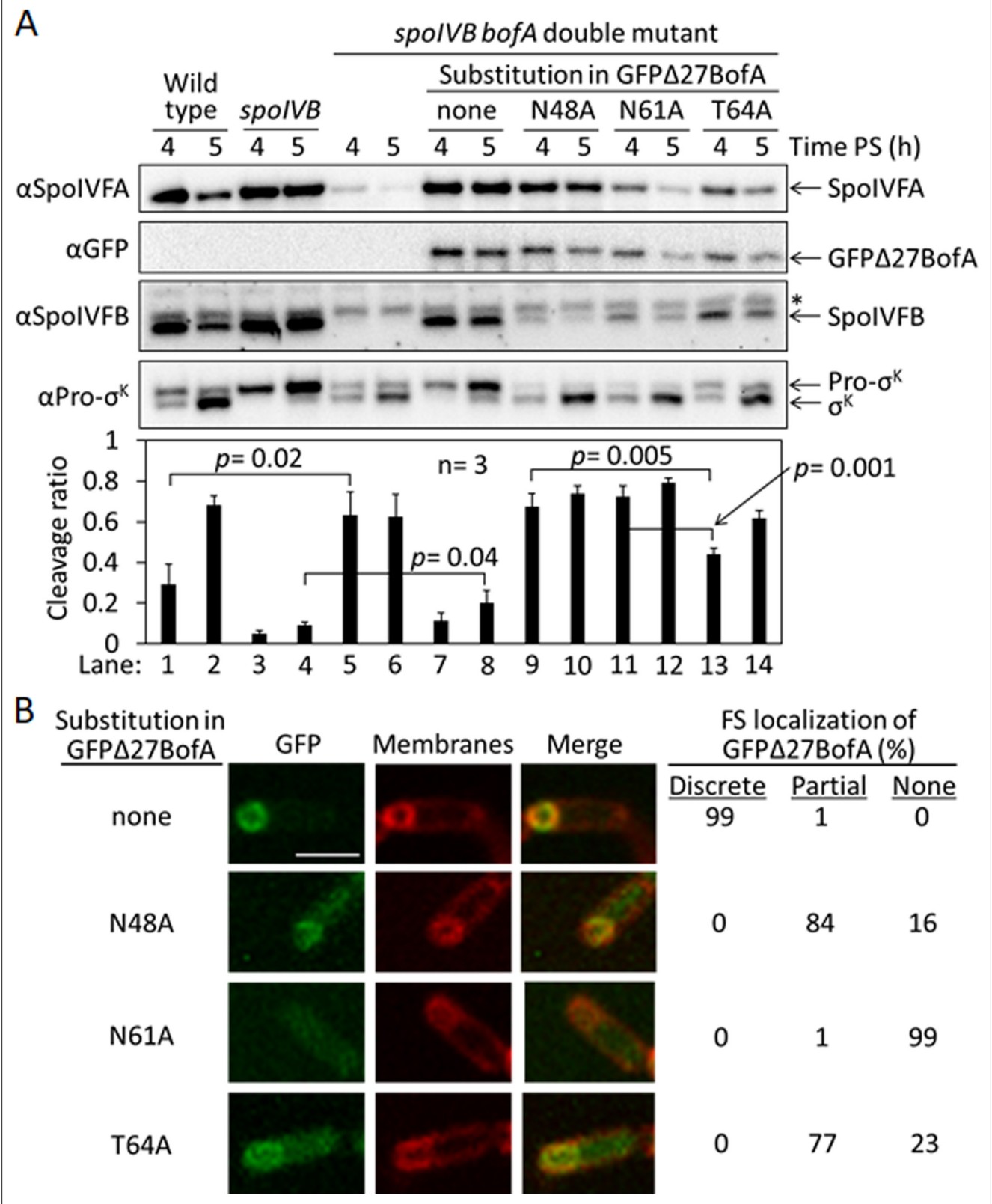

**Figure 3.** Effects of alanine substitutions in GFPΔ27BofA during *Bacillus subtilis* sporulation. (**A**) Effects of GFPΔ27BofA variants (N48A, N61A, and T64A) on Pro-σ$^K$ cleavage. Wild-type strain PY79, a *spoIVB165* null mutant, a *spoIVB165 bofA::erm* double mutant, and the double mutant with P$_{bofA}$-*gfpΔ27bofA* integrated at *amyE* to express GFPΔ27BofA with no substitution (none) or the indicated Ala substitution, were starved to induce sporulation. Samples collected at 4 and 5 hr poststarvation (PS) were subjected to immunoblot analysis with antibodies against SpoIVFA, GFP,

*Figure 3 continued on next page*

*Figure 3 continued*

SpoIVFB, and Pro-σ$^K$. The graph shows quantification of the cleavage ratio [σ$^K$/(Pro-σ$^K$+σ$^K$)] for three biological replicates. Error bars, 1 standard deviation. Student's two-tailed t-tests were performed to compare certain cleavage ratios (p values). (**B**) Localization of GFPΔ27BofA and the three variants. Samples collected at 3 hr PS were treated with FM 4–64 to stain membranes. Confocal microscopy images of fluorescence from GFPΔ27BofA, membranes, and merged images are shown for representative sporangia with discrete (no substitution in GFPΔ27BofA, designated 'none'), partial (N48A and T64A), or no forespore (FS) localization (N61A). Scale bar, 1 μm. The percentage of sporangia (44–93 counted; nonsporulating cells were not counted) with each localization pattern is shown.

The online version of this article includes the following source data and figure supplement(s) for figure 3:

**Source data 1.** Immunoblot images (annotated) and quantification of cleavage assays (*Figure 3A*), and confocal microscopy images (see the readme file) and a table of the numbers of sporangia in which FS localization of GFPΔ27BofA and the three variants were counted (*Figure 3B*).

**Source data 2.** Immunoblot images (raw) (*Figure 3A*).

**Figure supplement 1.** GFPΔ27BofA variants are intact during *Bacillus subtilis* sporulation.

**Figure supplement 1—source data 1.** Immunoblot images (raw and annotated).

Since GFPΔ27BofA co-localizes with SpoIVFA and SpoIVFB to the outer FS membrane during sporulation (*Resnekov et al., 1996*; *Rudner et al., 2002*; *Rudner and Losick, 2002*), we examined the ability of the GFPΔ27BofA variants to localize to the FS. As a control, GFPΔ27BofA produced in the *spoIVB165 bofA::erm* double mutant localized discretely to the FS at 3 hr PS (*Figure 3B*). The N48A and T64A variants localized partially to the FS, but the MC cytoplasm also exhibited GFP fluorescence, suggesting partial mislocalization. The N61A variant failed to localize to the FS, instead showing GFP fluorescence throughout the MC cytoplasm. The fusion proteins were intact, demonstrating that the cytoplasmic GFP fluorescence was not attributable to breakdown (*Figure 3—figure supplement 1*). Inability of the N61A variant to localize to the FS may explain the loss of SpoIVFB inhibition and the abundant cleavage of Pro-σ$^K$ at 4 hr (*Figure 3A*, lane 11). Importantly, the similar ability of the N48A and T64A variants to localize to the FS (*Figure 3B*) does not account for their differential effects on the level of SpoIVFB and its ability to cleave Pro-σ$^K$ (*Figure 3A*). The strain producing GFPΔ27BofA N48A exhibited less SpoIVFB yet more Pro-σ$^K$ cleavage at 4 hr (lane 9) than the strain producing GFPΔ27BofA T64A (lane 13), so the N48A substitution more severely impairs the ability of GFPΔ27BofA to inhibit SpoIVFB.

## BofA TMS2 occupies the SpoIVFB active site cleft

We hypothesized that TMS2 of BofA occupies the SpoIVFB active site cleft in the inhibition complex, because the GFPΔ27BofA N48 side chain is important for inhibition (*Figures 2 and 3*) and located near the middle of predicted TMS2 (*Figure 2—figure supplement 1*). To begin testing our hypothesis, we devised a strategy based on a model of the SpoIVFB membrane domain derived from the crystal structure of an archaeal homolog (*Feng et al., 2007*) and supported by cross-linking studies of catalytically inactive SpoIVFB E44Q in complex with Pro-σ$^K$(1–127), whose Proregion appears to occupy the SpoIVFB active site cleft (*Zhang et al., 2013*; *Halder et al., 2017*; *Zhang et al., 2016*). A cleft between TMS1 and TMS6 of SpoIVFB may gate substrate access to the active site (*Ramirez-Guadiana et al., 2018*) formed by a zinc ion near E44 of the HELGH metalloprotease motif within SpoIVFB TMS2 (*Rudner et al., 1999*, *Yu and Kroos, 2000*; *Figure 4—figure supplement 1A*). We reasoned that BofA TMS2 occupancy of the SpoIVFB active site cleft (*Figure 4—figure supplement 1B*) would exclude the substrate Proregion and may be detectable using a disulfide cross-linking approach. To implement disulfide cross-linking in the context of the inhibition complex formed with pET Quartet plasmids in *E. coli*, two of the proteins must have a single-Cys residue at positions hypothesized to be in proximity and the other two proteins must be Cys-less.

We took advantage of the sole Cys residue of BofA, C46, near N48 and the middle of predicted TMS2 (*Figure 2—figure supplement 1*), to test whether BofA C46 could be cross-linked to SpoIVFB E44C (since E44 is at the active site), using appropriate protein variants. Single-Cys E44C cytTM-SpoIVFB (which is catalytically inactive) and Cys-less Pro-σ$^K$(1–127) (which can be cleaved by active SpoIVFB) have been used previously (*Zhang et al., 2013*). We created functional variants of Cys-less SpoIVFA and single-Cys MBPΔ27BofA (GFP has Cys residues but MBP lacks them, leaving BofA C46 as the sole Cys residue) (*Figure 4—figure supplement 2*). As a negative control, we replaced C46 with Ser to obtain functional Cys-less MBPΔ27BofA. We induced *E. coli* to coproduce combinations

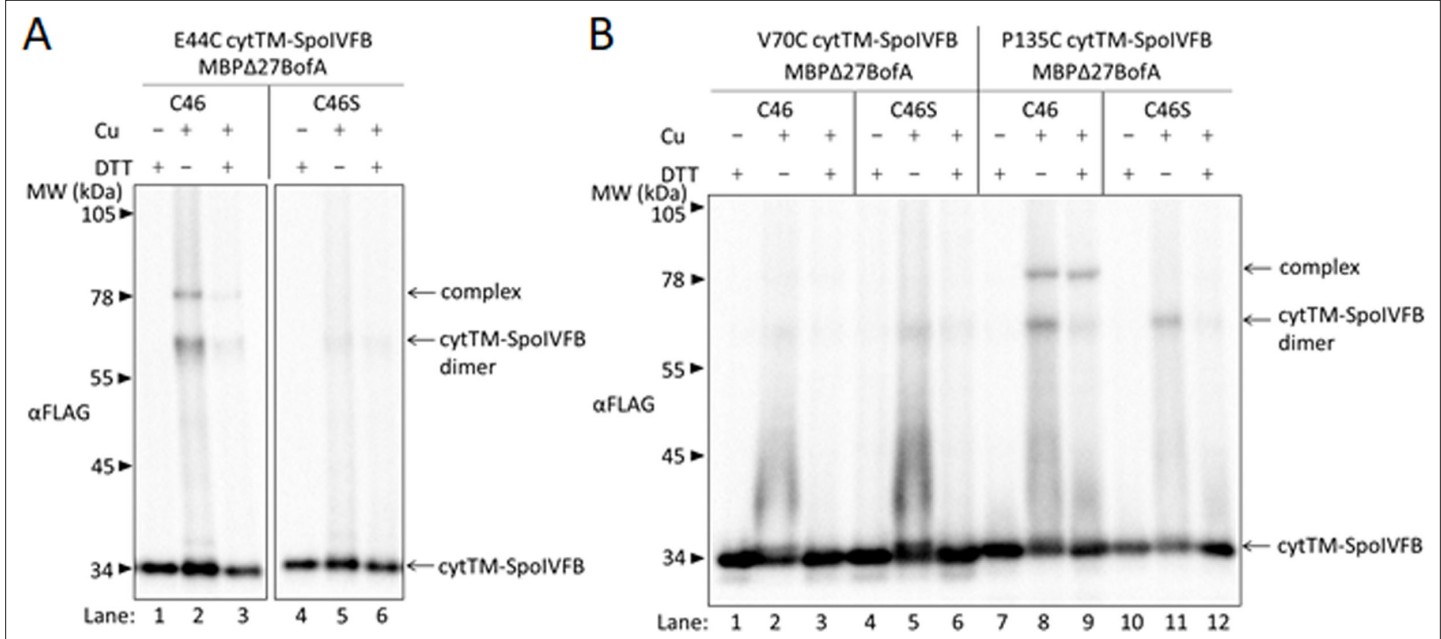

**Figure 4.** BofA TMS2 is near the SpoIVFB active site. (**A**) Disulfide cross-linking of E44C at the cytTM-SpoIVFB active site to C46 in TMS2 of MBPΔ27BofA. pET Quartet plasmids were used to produce single-Cys E44C cytTM-SpoIVFB in combination with MBPΔ27BofA C46 (pSO91) or Cys-less MBPΔ27BofA C46S as a negative control (pSO110), and Cys-less variants of SpoIVFA and Pro-σ^K(1–127) in *Escherichia coli*. Samples collected after 2 hr of IPTG induction were treated for 60 min with Cu$^{2+}$(phenanthroline)$_3$ (Cu+) to promote disulfide bond formation or with 2-phenanthroline (Cu–) as a negative control, then treated with TCA to precipitate proteins and resuspended in sample buffer with DTT (+) to reverse cross-links or without (–) to preserve cross-links, and finally subjected to immunoblot analysis with FLAG antibodies to visualize cytTM-SpoIVFB monomer, dimer, and complex with MBPΔ27BofA. (**B**) Disulfide cross-linking of V70C or P135C near the cytTM-SpoIVFB active site to C46 in TMS2 MBPΔ27BofA. pET Quartet plasmids were used to produce single-Cys V70C or P135C cytTM-SpoIVFB E44Q variants in combination with MBPΔ27BofA (pSO92 and pSO93) or Cys-less MBPΔ27BofA C46S as a negative control (pSO111 and pSO112), and Cys-less variants of SpoIVFA and Pro-σ^K(1–127) in *E. coli*. Samples collected after 2 hr of IPTG induction were treated and subjected to immunoblot analysis as in (**A**). A representative result from at least two biological replicates is shown in (**A**) and (**B**).

The online version of this article includes the following source data and figure supplement(s) for figure 4:

**Source data 1.** Immunoblot images (raw and annotated).

**Figure supplement 1.** Models of SpoIVFB and BofA TMS2.

**Figure supplement 1—source data 1.** PyMOL session file used to produce the images.

**Figure supplement 2.** Cys-less variants of SpoIVFA and MBPΔ27BofA inhibit cleavage of Pro-σ^K(1–127) by cytTM-SpoIVFB in *Escherichia coli*.

**Figure supplement 2—source data 1.** Immunoblot images (raw and annotated) and quantification of cleavage assays.

**Figure supplement 3.** BofA TMS2 is in proximity to the active site of SpoIVFB.

**Figure supplement 4.** Full-length BofA is in proximity to the active site of SpoIVFB.

**Figure supplement 4—source data 1.** Immunoblot images (raw and annotated).

**Figure supplement 5.** BofA TMS2 has a preferred orientation in the active site cleft of SpoIVFB.

**Figure supplement 5—source data 1.** Immunoblot images (raw and annotated) (*Figure 4—figure supplement 5B and C*) and quantification of cleavage assays (*Figure 4—figure supplement 5B*).

of four proteins and treated cells with the oxidant Cu$^{2+}$(phenanthroline)$_3$ to promote disulfide bond formation.

For MBPΔ27BofA C46 (*Figure 4A*, lane 2), but not the Cys-less C46S negative control (lane 5), treatment with oxidant caused formation of a species of the expected size for a cross-linked complex with single-Cys E44C cytTM-SpoIVFB, as detected by immunoblotting with anti-FLAG antibodies. Treatment with the reducing agent DTT greatly diminished the abundance of the apparent complex, consistent with cross-link reversal (lane 3). We also observed a species of the expected size for a cross-linked dimer of single-Cys E44C cytTM-SpoIVFB. Formation of the apparent dimer varied, as reported previously (*Zhang et al., 2013*). As expected, anti-MBP antibodies detected the presumptive cross-linked

complex of MBPΔ27BofA C46 with single-Cys E44C cytTM-SpoIVFB, albeit weakly, and the negative control with E44Q rather than E44C failed to form the complex (*Figure 4—figure supplement 3*, lanes 2 and 5). Since the signal for the complex was stronger with anti-FLAG antibodies (*Figure 4A*), we used those antibodies in the cross-linking experiments reported below.

In addition to E44 of SpoIVFB, V70 in a predicted membrane-reentrant loop and P135 in a predicted short loop interrupting TMS4 (*Figure 1A* and *Figure 4—figure supplement 1A*) were shown to be in proximity to the Proregion of Pro-σ$^K$(1–127) (*Zhang et al., 2013*). Therefore, we tested cross-linking of MBPΔ27BofA C46 to single-Cys V70C or P135C cytTM-SpoIVFB E44Q variants. We included the inactivating E44Q substitution since the V70C and P135C variants (unlike the E44C variant) could cleave Cys-less Pro-σ$^K$(1–127) (*Zhang et al., 2013*), even though we expected the inhibitory proteins to almost completely inhibit cleavage (*Figure 4—figure supplement 2*). MBPΔ27BofA C46 formed a complex with the P135C variant, but not with the V70C variant (*Figure 4B*, lanes 2 and 8). As expected, Cys-less MBPΔ27BofA C46S failed to form a complex with either variant (lanes 5 and 11). Full-length BofA C46 (lacking MBP) also formed a complex of the expected (smaller) size with the E44C and P135C variants, but not with the V70C variant (*Figure 4—figure supplement 4*, lanes 2, 14, and 8). As expected, Cys-less BofA C46S failed to form a complex with any of the variants (lanes 5, 11, and 17). Our cross-linking results show that BofA TMS2 occupies the SpoIVFB active site cleft in the inhibition complex, placing BofA C46 in proximity to SpoIVFB E44 and P135, but not V70.

Based on our initial cross-linking results, we modeled BofA TMS2 in the SpoIVFB active site cleft, and tested predictions of the model using additional disulfide cross-linking experiments (*Figure 4—figure supplement 5*). The results confirmed predictions of the initial model and suggested a preferred orientation of BofA TMS2 in the SpoIVFB active site cleft, which led to the refined model shown in *Figure 4—figure supplement 1B*.

## BofA and SpoIVFA do not prevent Pro-σ$^K$(1–127) from interacting with SpoIVFB

Since our cross-linking results show that BofA TMS2 occupies the SpoIVFB active site cleft, we tested whether BofA and SpoIVFA prevent Pro-σ$^K$(1–127) from interacting with SpoIVFB in *E. coli*. We coproduced a catalytically inactive E44C cytTM-SpoIVFB variant with a FLAG$_2$ epitope tag with Pro-σ$^K$(1–127), SpoIVFA, and GFPΔ27BofA. We then prepared cell lysates and detergent-solubilized proteins, which we co-immunoprecipitated in pull-down assays with anti-FLAG antibody beads. We observed all four proteins in the bound sample (*Figure 5A*, lane 4), so GFPΔ27BofA and SpoIVFA did not completely prevent Pro-σ$^K$(1–127) from interacting with the cytTM-SpoIVFB variant. However, we detected only the cytTM-SpoIVFB variant when the bound sample was diluted tenfold (lane 3) to match the input sample concentration (lane 1), and we observed portions of the other three proteins in the unbound sample (lane 2), indicating that co-purification was inefficient. A negative control with the cytTM-SpoIVFB variant lacking the FLAG$_2$ epitope tag showed no Pro-σ$^K$(1–127) in the bound sample, but did show small amounts of GFPΔ27BofA and SpoIVFA (lane 8), indicative of weak, nonspecific binding to the beads.

We also performed pull-down assays with cobalt resin, which binds to the His$_6$ tag on Pro-σ$^K$(1–127), and we performed both types of pull-down assays (i.e., anti-FLAG antibody beads and cobalt resin) on proteins from *E. coli* coproducing full-length BofA rather than GFPΔ27BofA with the other three proteins (*Figure 5—figure supplement 1*). Neither BofA nor GFPΔ27BofA when coproduced with SpoIVFA prevented Pro-σ$^K$(1–127) from interacting with the cytTM-SpoIVFB variant. However, GFPΔ27BofA and SpoIVFA reduced co-purification of full-length Pro-σ$^K$-His$_6$ with the cytTM-SpoIVFB variant in both types of pull-down assays (*Figure 5—figure supplement 2*), as compared with Pro-σ$^K$(1–127) (*Figure 5A* and *Figure 5—figure supplement 1*), suggesting that the C-terminal half of Pro-σ$^K$ affects complex formation.

To explain the presence of substrate in the pulled-down protein complexes, but the absence of substrate cleavage (*Figure 1B*), we hypothesized that inhibitory proteins block substrate access to the SpoIVFB active site cleft, but do not prevent substrate interaction with the soluble C-terminal CBS domain of SpoIVFB (*Figure 1A*). The model of catalytically inactive SpoIVFB E44Q in complex with Pro-σ$^K$(1–127) predicts extensive interactions between the SpoIVFB CBS domain and the Pro-σ$^K$(1–127) C-terminal region (*Halder et al., 2017*). We used the model to guide the testing of single-Cys variants of the two proteins for disulfide cross-link formation. We discovered that A231C in the

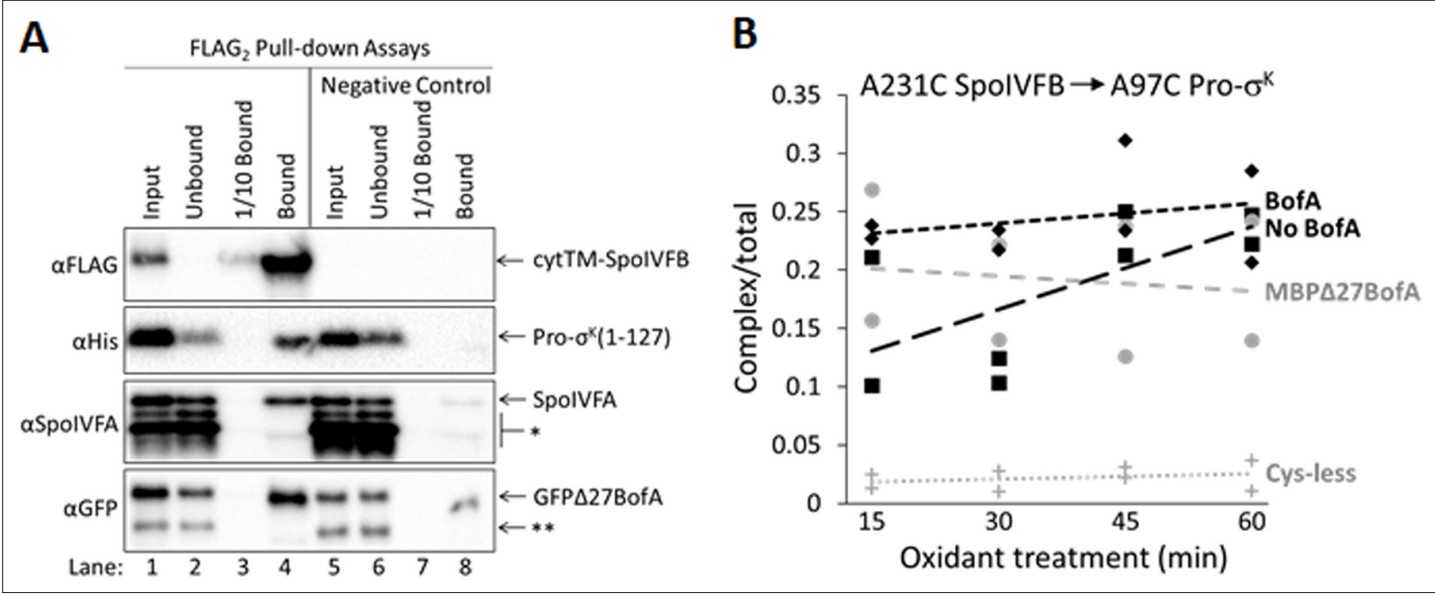

**Figure 5.** Inhibitory proteins do not prevent Pro-σ$^K$(1–127) from interacting with SpoIVFB. (**A**) Pro-σ$^K$(1–127), SpoIVFA, and GFPΔ27BofA co-purify with cytTM-SpoIVFB. pET Quartet plasmids were used to produce a catalytically inactive E44C cytTM-SpoIVFB variant with a FLAG$_2$ tag (pSO73), or a variant lacking FLAG$_2$ as a negative control (pSO149), in combination with Pro-σ$^K$(1–127), SpoIVFA, and GFPΔ27BofA in *Escherichia coli*. Samples collected after 2 hr of IPTG induction were subjected to co-immunoprecipitation with anti-FLAG antibody beads. Input, unbound, 1/10 bound (diluted to match input), and (undiluted) bound samples were subjected to immunoblot analysis with FLAG, penta-His, SpoIVFA, and GFP antibodies. The single star (*) indicates cross-reacting proteins below SpoIVFA and the double star (**) indicates a cross-reacting protein or breakdown species of GFPΔ27BofA that fail to co-purify. A representative result from two biological replicates is shown. (**B**) A231C in the cytTM-SpoIVFB CBS domain forms disulfide cross-links with A97C in the C-terminal region of Pro-σ$^K$(1–127) in the absence or presence of inhibitory proteins. pET Duet plasmids were used to produce single-Cys A231C cytTM-SpoIVFB E44Q in combination with single-Cys A97C Pro-σ$^K$(1–127) (pSO130; squares and line with long dashes labeled 'No BofA', although SpoIVFA is also absent) or with Cys-less Pro-σ$^K$(1–127) as a negative control (pSO255; crosses and line labeled 'Cys-less') in *E. coli*. pET Quartet plasmids were used to produce single-Cys A231C cytTM-SpoIVFB E44Q, single-Cys A97C Pro-σ$^K$(1–127), and Cys-less SpoIVFA in combination with Cys-less MBPΔ27BofA (pSO133; circles and line labeled 'MBPΔ27BofA') or with Cys-less full-length BofA (pSO246; diamonds and line labeled 'BofA') in *E. coli*. Samples collected after 2 hr of IPTG induction were treated for 15, 30, 45, or 60 min with Cu$^{2+}$(phenanthroline)$_3$ oxidant to promote disulfide bond formation and subjected to immunoblot analysis with FLAG antibodies to visualize the cytTM-SpoIVFB monomer, dimer, and complex with Pro-σ$^K$(1–127) (*Figure 5—figure supplement 3C*). Abundance of the complex was divided by the total amount of cytTM-SpoIVFB monomer, dimer, and complex. The ratio over time was plotted (n=2) with a best-fit trend line.

The online version of this article includes the following source data and figure supplement(s) for figure 5:

**Source data 1.** Immunoblot images (raw and annotated) (*Figure 5A*) and quantification of cross-linking (*Figure 5B*).

**Figure supplement 1.** Neither GFPΔ27BofA nor full-length BofA when coproduced with SpoIVFA prevent Pro-σ$^K$(1–127) from interacting with SpoIVFB.

**Figure supplement 1—source data 1.** Immunoblot images (raw and annotated).

**Figure supplement 2.** GFPΔ27BofA and SpoIVFA do not prevent full-length Pro-σ$^K$ from interacting with SpoIVFB.

**Figure supplement 2—source data 1.** Immunoblot images (raw and annotated).

**Figure supplement 3.** Disulfide cross-linking between the cytTM-SpoIVFB CBS domain and the Pro-σ$^K$(1–127) C-terminal region.

**Figure supplement 3—source data 1.** Immunoblot images (raw and annotated) and quantification of cleavage assays (*Figure 5—figure supplement 3B*).

cytTM-SpoIVFB CBS domain can be cross-linked to A97C in the Pro-σ$^K$(1–127) C-terminal region upon oxidant treatment of *E. coli* coproducing the proteins (*Figure 5—figure supplement 3A*). We showed that the Cys substitutions do not impair cytTM-SpoIVFB activity or Pro-σ$^K$(1–127) susceptibility to cleavage (*Figure 5—figure supplement 3B*). Finally, we measured time-dependent cross-linking in the presence or absence of Cys-less inhibitory proteins. Coproduction of inhibitory proteins had little or no effect on the formation of cross-linked complex (*Figure 5B* and *Figure 5—figure supplement 3C*). These results suggest that neither full-length BofA nor MBPΔ27BofA, when coproduced with SpoIVFA, prevent Pro-σ$^K$(1–127) from interacting with the CBS domain of SpoIVFB in *E. coli*, consistent with the results of our pull-down assays (*Figure 5A* and *Figure 5—figure supplement 1*).

## Inhibitory proteins block access of the substrate N-terminal Proregion to the SpoIVFB active site

SpoIVFB cleaves Pro-σ$^K$ (*Kroos et al., 1989*) and Pro-σ$^K$(1–127) (*Zhou and Kroos, 2004*) between residues S21 and Y22. In disulfide cross-linking experiments, Cys substitutions for several residues near the cleavage site in otherwise Cys-less Pro-σ$^K$(1–127) formed a cross-linked complex with single-Cys (E44C, V70C, or P135C) cytTM-SpoIVFB variants (*Zhang et al., 2013*). The complex was most abundant with the E44C and V70C variants, so we compared these interactions in the presence or absence of Cys-less inhibitory proteins.

SpoIVFB E44 is presumed to activate a water molecule for substrate peptide bond hydrolysis at the enzyme active site (*Rudner et al., 1999*; *Yu and Kroos, 2000*). To test access of the substrate Proregion to the enzyme active site, we first measured time-dependent cross-linking between single-Cys E44C cytTM-SpoIVFB and single-Cys (F18C, V20C, S21C, or K24C) Pro-σ$^K$(1–127) variants coproduced in the absence of inhibitory proteins in *E. coli*. The V20C and K24C Pro-σ$^K$(1–127) variants formed abundant complex that increased over time, but the F18C and S21C variants formed much less complex, only slightly more than the Cys-less Pro-σ$^K$(1–127) negative control (*Figure 6A* and *Figure 6—figure supplement 1A*). *Figure 6B* shows a representative immunoblot (60 min oxidant treatment). Upon coproduction with Cys-less MBPΔ27BofA and SpoIVFA, the V20C and K24C Pro-σ$^K$(1–127) variants formed much less complex and its abundance did not increase over time (*Figure 6C* and *Figure 6—figure supplement 1B*). *Figure 6D* shows a representative immunoblot (60 min oxidant treatment) for comparison with *Figure 6B*. Upon coproduction with Cys-less full-length BofA and SpoIVFA, abundance of the complex decreased similarly (*Figure 6—figure supplement 2*). *Figure 6E and F* summarize the cross-linking time courses with and without inhibitory proteins for the V20C and K24C Pro-σ$^K$(1–127) variants. The effects of MBPΔ27BofA (lacking TMS1) and full-length BofA were indistinguishable. These results likely explain why BofA TMS1 is dispensable for most of the inhibitory function of BofA (*Rudner and Losick, 2002*; *Zhou and Kroos, 2004*; *Figures 1B and 3A*). We conclude that inhibitory proteins block access of the substrate N-terminal Proregion to the SpoIVFB active site.

SpoIVFB V70 is located in a predicted membrane-reentrant loop (*Figure 1A*), which may bind to the Proregion and present it to the active site for cleavage (*Halder et al., 2017*) based on a study of *E. coli* RseP (*Akiyama et al., 2015*). Single-Cys V70C cytTM-SpoIVFB E44Q formed abundant cross-linked complex with F18C and K24C Pro-σ$^K$(1–127) variants in the absence of inhibitory proteins (*Zhang et al., 2013*; *Figure 6G and H*; *Figure 6—figure supplement 3A and B*). Upon coproduction with Cys-less MBPΔ27BofA and SpoIVFA, less complex formed (*Figure 6G and H*; *Figure 6—figure supplement 3C and D*). Coproduction with Cys-less full-length BofA and SpoIVFA further decreased complex formation (*Figure 6G and H*; *Figure 6—figure supplement 3E and F*). Since full-length BofA hindered cross-linking more than MBPΔ27BofA (lacking TMS1) (*Figure 6G and H*), both TMSs of BofA appear to interfere with the normal interaction between the SpoIVFB membrane-reentrant loop and the substrate Proregion. BofA may cause the SpoIVFB membrane-reentrant loop to be exposed since we observed four novel species (*Figure 6—figure supplement 3G*), perhaps due to cross-linking of the V70C cytTM-SpoIVFB variant to *E. coli* proteins.

Since full-length BofA also inhibited cleavage of Pro-σ$^K$(1–127) in *E. coli* (*Figure 1B*, longer exposure) and Pro-σ$^K$ in *B. subtilis* (*Figure 3A*, lanes 4 and 8) slightly more than GFPΔ27BofA (lacking TMS1), we compared time-dependent cross-linking of C46 of BofA and MBPΔ27BofA to single-Cys E44C and P135C cytTM-SpoIVFB variants. The results suggest that full-length BofA forms slightly more complex over time, whereas complex abundance did not increase over time with MBPΔ27BofA (*Figure 6—figure supplement 4*). Hence, BofA TMS1 may slightly enhance TMS2 occupancy in the SpoIVFB active site cleft.

To examine the extent to which inhibitory proteins hinder the interaction of the substrate with SpoIVFB, we used the model of catalytically inactive SpoIVFB E44Q in complex with Pro-σ$^K$(1–127) (*Halder et al., 2017*) to guide testing for disulfide cross-link formation and found that A214C in the cytTM-SpoIVFB linker (*Figure 1A*) can be cross-linked to A41C in Pro-σ$^K$(1–127) (*Figure 6—figure supplement 5A*). We showed that the Cys substitutions do not impair cytTM-SpoIVFB activity or Pro-σ$^K$(1–127) susceptibility to cleavage (*Figure 5—figure supplement 3B*). Finally, we measured time-dependent cross-linking in the presence or absence of Cys-less inhibitory proteins. Interestingly, full-length BofA hindered cross-linking more than MBPΔ27BofA (*Figure 6I* and *Figure 6—figure supplement 5B*), similar to cross-linking between V70C in the cytTM-SpoIVFB membrane-reentrant

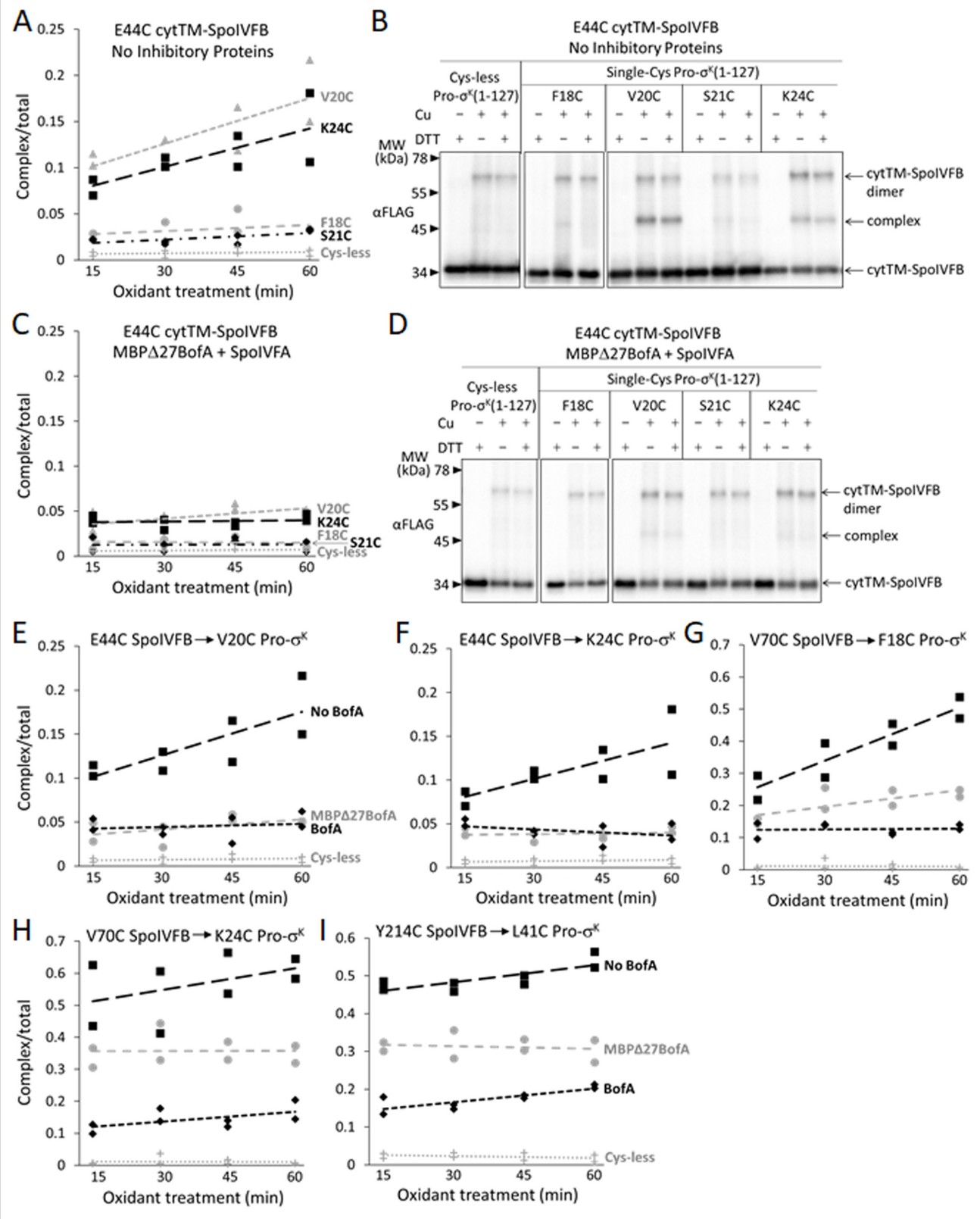

**Figure 6.** Inhibitory proteins block access of the Pro-σ^K(1–127) N-terminal region to the SpoIVFB active site and hinder interaction with the SpoIVFB membrane-reentrant loop and interdomain linker. (**A**) E44C at the cytTM-SpoIVFB active site forms abundant disulfide cross-links with V20C or K24C in the N-terminal region of Pro-σ^K(1–127) in the absence of inhibitory proteins. pET Duet plasmids were used to produce single-Cys E44C cytTM-SpoIVFB in combination with single-Cys F18C (pSO167), V20C (pSO169), S21C (pSO170), or K24C (pSO128) Pro-σ^K(1–127), or with Cys-less Pro-σ^K(1–127)

*Figure 6 continued on next page*

*Figure 6 continued*

(pSO79) as a negative control, in *Escherichia coli*. Samples collected after 2 hr of IPTG induction were treated with $Cu^{2+}$(phenanthroline)$_3$ oxidant for 15, 30, 45, or 60 min to promote disulfide bond formation and subjected to immunoblot analysis with FLAG antibodies to visualize the cytTM-SpoIVFB monomer, dimer, and complex with Pro-σ$^K$(1–127) (*Figure 6—figure supplement 1A*). Abundance of the complex was divided by the total amount of cytTM-SpoIVFB monomer, dimer, and complex. The ratio over time was plotted (n=2) with a best-fit trend line. (**B**) Representative immunoblots of 60 min samples from the experiment are described in (**A**). (**C**) MBPΔ27BofA and SpoIVFA decrease cross-linking between E44C cytTM-SpoIVFB and V20C or K24C Pro-σ$^K$(1–127). pET Quartet plasmids were used to produce single-Cys E44C cytTM-SpoIVFB in combination with single-Cys F18C (pSO163), V20C (pSO165), S21C (pSO166), or K24C (pSO131) Pro-σ$^K$(1–127), or with Cys-less Pro-σ$^K$(1–127) (pSO110) as a negative control, and Cys-less variants of MBPΔ27BofA and SpoIVFA in *E. coli*. Samples collected after 2 hr of IPTG induction were treated and subjected to immunoblot analysis as in (**A**) (*Figure 6—figure supplement 1B*). The complex/total ratio was plotted as in (**A**). (**D**) Representative immunoblots of 60 min samples from the experiment are described in (**C**). (**E, F**) Summaries of the effects of inhibitory proteins on cross-linking between E44C cytTM-SpoIVFB and V20C or K24C Pro-σ$^K$(1–127). Data from (**A**) (labeled 'No BofA' in (**E**), although SpoIVFA is also absent), (**C**) (labeled 'MBPΔ27BofA' in (**E**), although SpoIVFA is also present), and *Figure 6—figure supplement 2* (labeled 'BofA' in (**E**), although SpoIVFA is also present) are plotted along with Cys-less Pro-σ$^K$(1–127) as a negative control. In (**F**), symbols and lines are as in (**E**). (**G, H**) Summaries of the effects of inhibitory proteins on cross-linking between V70C in the cytTM-SpoIVFB membrane-reentrant loop and F18C or K24C in the Pro-σ$^K$(1–127) N-terminal region. Data from *Figure 6—figure supplement 3* are plotted using symbols and lines as in (**E**). (**I**) Summary of the effects of inhibitory proteins on cross-linking between Y214C in the cytTM-SpoIVFB interdomain linker and L41C in the Pro-σ$^K$(1–127) N-terminal region. Data from *Figure 6—figure supplement 5* are plotted using symbols and lines as in (**E**).

The online version of this article includes the following source data and figure supplement(s) for figure 6:

**Source data 1.** Quantification of cross-linking (*Figure 6A, C, and E–I*).

**Figure supplement 1.** Disulfide cross-linking between E44C at the active site of cytTM-SpoIVFB and the Proregion of Pro-σ$^K$(1–127) in the absence of inhibitory proteins or in the presence of MBPΔ27BofA and SpoIVFA.

**Figure supplement 1—source data 1.** Immunoblot images (raw and annotated) (*Figure 6—figure supplement 1A*).

**Figure supplement 1—source data 2.** Immunoblot images (raw and annotated) (*Figure 6—figure supplement 1B*).

**Figure supplement 2.** Full-length BofA and SpoIVFA decrease cross-linking between E44C cytTM-SpoIVFB and V20C or K24C Pro-σ$^K$(1–127).

**Figure supplement 2—source data 1.** Immunoblot images (raw and annotated) (*Figure 6—figure supplement 2A*) and quantification of cross-linking (*Figure 6—figure supplement 2B*).

**Figure supplement 3.** Disulfide cross-linking between V70C in the cytTM-SpoIVFB membrane-reentrant loop and F18C or K24C in the Pro-σ$^K$(1–127) N-terminal region is decreased more by full-length BofA than by MBPΔ27BofA (lacking TMS1).

**Figure supplement 3—source data 1.** Immunoblot images (raw and annotated) (*Figure 6—figure supplement 3A, C, E, and G*) and quantification of cross-linking (*Figure 6—figure supplement 3B, D, and F*).

**Figure supplement 4.** Comparison of disulfide cross-linking between C46 in TMS2 of full-length BofA or MBPΔ27BofA (lacking TMS1) and E44C at or P135C near the active site of cytTM-SpoIVFB.

**Figure supplement 4—source data 1.** Immunoblot images (raw and annotated) (*Figure 6—figure supplement 4A, B, D, and E*) and quantification of cross-linking (*Figure 6—figure supplement 4C and F*).

**Figure supplement 5.** Disulfide cross-linking between Y214C in the cytTM-SpoIVFB interdomain linker and L41C in the Pro-σ$^K$(1–127) N-terminal region is decreased more by full-length BofA than by MBPΔ27BofA (lacking TMS1).

**Figure supplement 5—source data 1.** Immunoblot images (raw and annotated).

loop and F18C or K24C near the cleavage site in Pro-σ$^K$(1–127) (*Figure 6G and H*). The similar pattern suggests that in both cases BofA TMS2 partially interferes with the interaction and BofA TMS1 augments the interference.

## A model of SpoIVFB in complex with BofA and parts of SpoIVFA and Pro-σ$^K$

We generated computational models using a similar protocol as described previously (*Halder et al., 2017*), but included additional constraints reflecting experimental cross-linking data reported herein, as well as newly predicted intra- and inter-chain contacts based on co-evolutionary couplings. The two final models are: (1) full-length SpoIVFB modeled as a tetramer, with part of one Pro-σ$^K$ molecule (residues 1–114), referred to as 'fb.sigk'; (2) full-length SpoIVFB, again modeled as a tetramer, with one molecule each of full-length BofA and parts of Pro-σ$^K$ (residues 38–114) and SpoIVFA (residues 65–111), referred to as 'fb.sigk.bofa.fa.' We omitted residues of Pro-σ$^K$ and SpoIVFA that we could not place with sufficient confidence. *Figure 7—figure supplement 1* shows the first model.

*Figure 7* illustrates relevant features of the second model (only one molecule of SpoIVFB is shown). The first side view shows SpoIVFB with BofA TMS2 and the C-terminal part of Pro-σ$^K$(1–127)

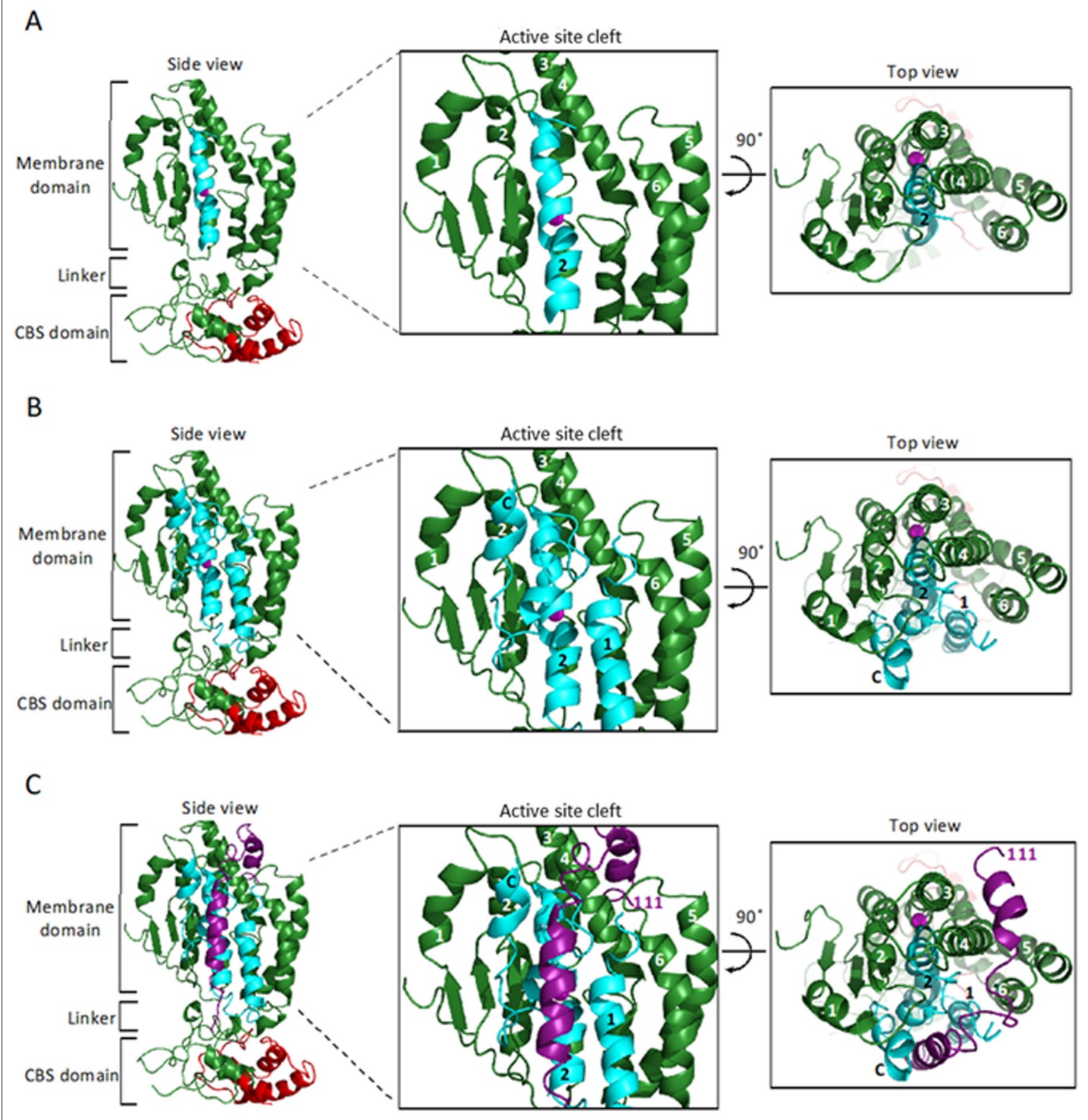

**Figure 7.** Model of SpoIVFB monomer with BofA and parts of SpoIVFA and Pro-σ$^K$. (**A**) Model of SpoIVFB, BofA TMS2, and the C-terminal part of Pro-σ$^K$(1–127). At *Left*, a side view of the complex, showing the six TMSs of the SpoIVFB membrane domain with the active site zinc ion (magenta), the interdomain linker, and the CBS domain (green), BofA TMS2 (cyan), and Pro-σ$^K$(38–114) (red). In the enlarged view of the active site cleft (*Center*), TMSs 1–6 of SpoIVFB and TMS2 of BofA are numbered. At *Right*, a top view is shown. (**B**) Model of SpoIVFB with full-length BofA and Pro-σ$^K$(38–114). Predicted TMSs 1 and 2 of BofA are numbered and its C-terminal region is labeled 'C' near the C-terminus in the views shown in the *Center* and at *Right*. (**C**) Model of SpoIVFB with full-length BofA, SpoIVFA(65–111) (purple, residue 111 is numbered), and Pro-σ$^K$(38–114). TMS, transmembrane segment.

The online version of this article includes the following source data and figure supplement(s) for figure 7:

*Figure 7 continued on next page*

*Figure 7 continued*

**Source data 1.** PyMOL session file used to produce the images and PDB file of the model of a SpoIVFB tetramer with BofA and parts of SpoIVFA and Pro-σ^K.

**Figure supplement 1.** Model of SpoIVFB tetramer with part of one Pro-σ^K molecule.

**Figure supplement 1—source data 1.** PyMOL session file used to produce the images and PDB file of the model of a SpoIVFB tetramer with part of one molecule of Pro-σ^K.

**Figure supplement 2.** Model of BofA showing conserved residues important for SpoIVFB inhibition.

**Figure supplement 2—source data 1.** PyMOL session file used to produce the images.

(*Figure 7A*). The membrane domain of SpoIVFB (green) interacts with BofA TMS2 (cyan) while the interdomain linker and CBS domain of SpoIVFB interact with the modeled portion of Pro-σ^K(1–127) (red). The enlarged view of the SpoIVFB active site cleft shows BofA TMS2 surrounded by SpoIVFB (TMSs labeled 1–6). The top view of the membrane domain emphasizes proximity between BofA TMS2, SpoIVFB TMSs 2–4, and the zinc ion (magenta) involved in catalysis.

The second side view shows SpoIVFB with full-length BofA and the C-terminal part of Pro-σ^K(1–127) (*Figure 7B*). Our model predicts that BofA contains two TMSs and a membrane-embedded C-terminal region (labeled C near the C-terminus in the enlarged view of the SpoIVFB active site cleft) that forms two short α-helices connected by a turn. The enlarged and top views show that BofA interacts extensively with SpoIVFB and occupies its active site cleft, which would sterically hinder access of the Proregion of Pro-σ^K. In the model, the conserved residue N48 in BofA TMS2 is near T64 and N61. Both T64 and N61 are in a loop that precedes the first short α-helix of the C-terminal region (*Figure 7—figure supplement 2*). All three residues are likely able to interact, but the constraints from experiments and co-evolutionary couplings are not sufficient for predicting the exact side-chain orientations with high certainty. So, even though the N61 side chain is shown pointing away from the N48 and T64 side chains in the model, slight structural rearrangements within the modeling constraints could allow the N61 side chain to interact more directly with the N48 and/or T64 side chain(s). Alternatively, the three conserved residues may contact other residues within BofA, based on co-evolutionary couplings, to stabilize the BofA structure or they may be essential for interactions with nearby elements of SpoIVFA and SpoIVFB.

*Figure 7C* shows the addition of the modeled portion of SpoIVFA (purple). Co-evolutionary couplings predict that SpoIVFA contacts the BofA C-terminal region and SpoIVFB TMS4. Hence, the model predicts that SpoIVFA stabilizes the interaction of BofA with SpoIVFB.

## Discussion

Our results provide evidence that BofA TMS2 occupies the SpoIVFB active site cleft. Both inhibitory proteins block access of the substrate N-terminal Proregion to the SpoIVFB active site, but do not prevent interaction between the C-terminal region of Pro-σ^K(1–127) and the SpoIVFB CBS domain. The mechanism of SpoIVFB inhibition is novel in comparison with previously known mechanisms of IP regulation. Structural modeling predicts that conserved BofA residues interact to stabilize TMS2 and a membrane-embedded C-terminal region. The model also predicts that SpoIVFA contacts the BofA C-terminal region and SpoIVFB TMS4, bridging the two proteins to stabilize the inhibition complex. The model has clear implications for relief of SpoIVFB and its orthologs from inhibition during sporulation, as well as for IP inhibitor design.

### A novel mechanism of IP regulation

Previous work revealed three mechanisms of IP regulation—substrate localization, substrate extramembrane domain cleavage, and substrate interaction with an adapter protein. Substrate localization to a different organelle than its cognate IP is common in eukaryotic cells (*Kühnle et al., 2019*; *Morohashi and Tomita, 2013*). An early example of this regulatory mechanism emerged from studies of S2P. SREBP substrates are retained in the ER by SCAP and Insig proteins, then transported to the Golgi when more cholesterol is needed (*Rawson, 2003*; *Figure 8A*).

RIP of SREBPs also involves extramembrane loop cleavage, referred to as site-1 cleavage, prior to site-2 cleavage by S2P (*Brown et al., 2000*; *Figure 8A*). Substrates of bacterial metallo IPs (with

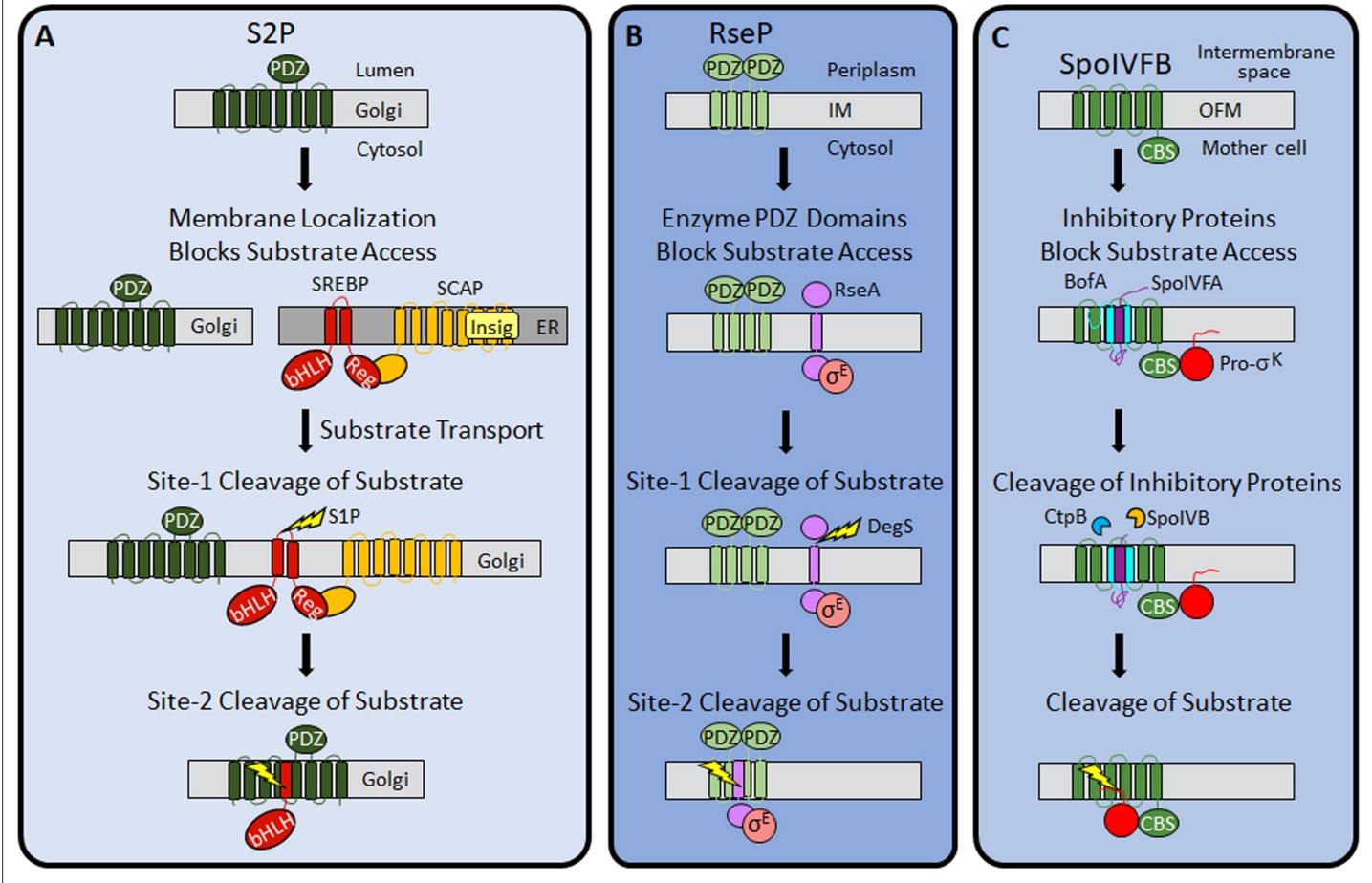

**Figure 8.** Regulation of intramembrane proteolysis by SpoIVFB differs from that of S2P and RseP. (**A**) S2P regulation. Human S2P localizes to membranes of the Golgi apparatus and has eight predicted TMSs and a PDZ domain (*Kroos and Akiyama, 2013*; *Ha, 2009*; *Zelenski et al., 1999*). SREBP substrates localize to endoplasmic reticulum (ER) membranes *via* interaction of their regulatory domain (Reg) with SCAP, which is bound by Insig proteins when the cholesterol level is high (*Rawson, 2013*). Cholesterol depletion releases SREBP-SCAP complexes from Insig and allows vesicular substrate transport to the Golgi membranes (*Nohturfft et al., 1999*; *DeBose-Boyd et al., 1999*; *Chen and Zhang, 2010*). Cleavage (lightning bolt) by S1P at site-1 in the luminal loop of SREBPs promotes separation of the TMSs and site-2 cleavage by S2P (*Sakai et al., 1996*; *Sakai et al., 1997*). S2P cleavage releases the basic helix-loop-helix (bHLH) domain of SREBP substrates into the cytosol, for subsequent entry into the nucleus and transcriptional activation of genes required for fatty acid and cholesterol synthesis and uptake (*Horton et al., 2003*). (**B**) RseP regulation. RseP localizes to the *Escherichia coli* inner membrane (IM) and has four predicted TMSs and tandem PDZ domains (*Kroos and Akiyama, 2013*; *Kanehara et al., 2001*; *Drew et al., 2002*). The substrate RseA also localizes to the IM and sequesters σ$^E$ in the absence of envelope stress (*Ades et al., 1999*). The PDZ domains of RseP block access of the RseA/σ$^E$ complex to the RseP active site (*Hizukuri et al., 2014*). Stress causes misfolding of outer membrane proteins and accumulation of lipopolysaccharide intermediates, which trigger site-1 cleavage by DegS in the RseA periplasmic domain (*Lima et al., 2013*; *Walsh et al., 2003*). Removal of the RseA extramembrane domain permits entry into the RseP active site cleft for site-2 cleavage (*Kanehara et al., 2002*; *Alba et al., 2002*). Release of RseA/σ$^E$ into the cytosol allows ClpXP protease to degrade RseA, liberating σ$^E$ to direct transcription of stress-responsive genes (*Flynn et al., 2003*; *Chaba et al., 2007*). (**C**) SpoIVFB regulation. SpoIVFB localizes to the outer forespore membrane (OFM) during *Bacillus subtilis* sporulation and has six predicted TMSs and a CBS domain (*Kroos and Akiyama, 2013*; *Cutting et al., 1991b*; *Resnekov et al., 1996*; *Kinch et al., 2006*). Our results support a novel regulatory mechanism in which the inhibitory proteins BofA and SpoIVFA occupy the SpoIVFB active site cleft and block access of the Pro-σ$^K$ Proregion. The inhibitory proteins did not prevent the interaction of Pro-σ$^K$ with the SpoIVFB CBS domain in our *E. coli* system. Proteases SpoIVB and CtpB are secreted from the forespore into the intermembrane space and cleave the inhibitory proteins (*Dong and Cutting, 2003*; *Campo and Rudner, 2006*; *Campo and Rudner, 2007*; *Zhou and Kroos, 2005*), allowing the Pro-σ$^K$ Proregion to enter the SpoIVFB active site cleft for cleavage (*Rudner et al., 1999*; *Yu and Kroos, 2000*; *Zhou et al., 2009*), which releases σ$^K$ into the mother cell to direct transcription of sporulation genes (*Kroos et al., 1989*; *Eichenberger et al., 2004*). TMS, transmembrane segment.

the exception of SpoIVFB) typically also require initial regulated site-1 cleavage by another protease (*Urban, 2009*). A well-studied example in *E. coli* involves envelope stress regulating site-1 cleavage of the RseA extramembrane domain by DegS, allowing site-2 cleavage by RseP (*Kroos and Akiyama, 2013*; *Lima et al., 2013*; *Figure 8B*). Substrates of aspartyl IPs likewise require an initial cleavage

by another protease, referred to as ectodomain shedding (*Lichtenthaler et al., 2018*; *Beard et al., 2019*). Shedding and site-1 cleavage of RseA allow substrate entry to the IP active site cleft. Without the initial cleavage, the substrate extramembrane domain prevents entry, due to size exclusion by the IP. In RseP, tandem PDZ domains act as a size-exclusion filter (*Hizukuri et al., 2014*; *Figure 8B*). The nicastrin subunit plays a similar role in the metazoan aspartyl IP γ-secretase (*Bolduc et al., 2015*).

Only two cases of IP regulation by substrate interaction with an adapter protein have been reported. In the bacterium *Mycobacterium tuberculosis*, the adapter protein Ppr1 bridges the metallo IP Rip1 to one of its substrates, but not to others (*Schneider and Glickman, 2013*). The γ-secretase subunits nicastrin and PEN-2 also appear to interact with substrate before passage to the active site (*Fukumori and Steiner, 2016*).

Our results support a novel mechanism of IP regulation—inhibitory proteins block substrate access to the SpoIVFB active site (*Figure 8C*). Many soluble proteases are regulated by inhibitory proteins, prodomains, or prosegments that sterically hinder substrate access to their active site (*Arolas et al., 2018*), but analogous regulation of an IP has not been reported previously. In addition to steric hindrance, inhibition of soluble metalloproteases often involves a residue of the inhibitory protein, prodomain, or prosegment with a side chain that coordinates the catalytic metal ion in place of a water molecule necessary for substrate peptide bond hydrolysis (*Arolas et al., 2018*; *Van Wart and Birkedal-Hansen, 1990*). *B. subtilis* BofA C46 is a potential zinc ligand near the SpoIVFB active site (*Figure 4* and *Figure 4—figure supplements 3 and 4*), but most BofA orthologs lack a Cys near the middle of predicted TMS2 (*Figure 2—figure supplement 1*). BofA orthologs have a highly conserved Asn (also a potential zinc ligand) at the position corresponding to N48 of *B. subtilis* BofA. N48 is crucial for SpoIVFB inhibition (*Figures 2 and 3*), but the N48 side chain points away from the zinc ion in our structural model (*Figure 7—figure supplement 2*). The model instead predicts that the N48 side chain may interact with the T64 side chain, another conserved BofA residue (*Figure 2—figure supplement 1*) shown to be important for SpoIVFB inhibition (*Figures 2 and 3*). Therefore, we favor a simple steric hindrance mechanism of inhibition not involving a BofA residue acting as a zinc ligand.

We propose that the BofA N48, N61, and T64 side chains are critical for interactions that stabilize a structural domain including TMS2 and a membrane-embedded C-terminal region, which in turn interact with SpoIVFA and SpoIVFB (*Figure 7C*), accounting for most of the steric hindrance mechanism of inhibition. This proposal can explain several experimental observations. First, Ala substitutions for N48, N61, and T64 in GFPΔ27BofA accumulated normally in *E. coli* in the absence of the other proteins (*Figure 2—figure supplement 2*), suggesting the variants fold almost normally. However, all three variants allowed Pro-σ^K(1–127) cleavage and reduced the SpoIVFA and cytTM-SpoIVFB levels upon coproduction (*Figure 2*), so subtle changes in the proposed BofA structural domain may alter formation of the inhibition complex and lead to protein instability. Similarly, all three GFPΔ27BofA variants accumulated normally at 4 hr during *B. subtilis* sporulation, but allowed premature cleavage of Pro-σ^K, despite reducing the level of SpoIVFB and in some cases SpoIVFA (*Figure 3A*), consistent with altered assembly of the inhibition complex. In agreement, the Ala substitutions caused partial or complete mislocalization of GFPΔ27BofA during sporulation (*Figure 3B*), which normally relies on SpoIVFA (*Rudner and Losick, 2002*).

A second observation that our proposed model for the SpoIVFB inhibition complex (*Figure 7C*) can explain is the requirement for the C-terminal end of BofA. Deletion of three residues from the BofA C-terminal end caused a loss of function or stability in *B. subtilis* (*Ricca et al., 1992*; *Varcamonti et al., 1997*). We found that deleting the three residues or changing them to Ala in GFPΔ27BofA caused loss of SpoIVFB inhibition in our *E. coli* system (*Figure 2—figure supplement 3A*). Our model predicts that the three BofA residues interact with SpoIVFA and SpoIVFB (*Figure 7C*), which can be tested by cross-linking experiments. Our model also predicts that SpoIVFA bridges the BofA C-terminal region and SpoIVFB TMS4 to stabilize the inhibition complex, which may explain the dependence of SpoIVFB inhibition on SpoIVFA and BofA (*Cutting et al., 1990*; *Cutting et al., 1991b*; *Ricca et al., 1992*) or GFPΔ27BofA (*Rudner and Losick, 2002*) during *B. subtilis* sporulation and on SpoIVFA and GFPΔ27BofA in *E. coli* (*Figure 1C*).

Third, our proposed model for the SpoIVFB inhibition complex (*Figure 7C*) can explain why GFPΔ27BofA inhibits substrate cleavage (*Figures 1B and 3A*) and MBPΔ27BofA occupies the SpoIVFB active site cleft (*Figure 6—figure supplement 4*) nearly as well as full-length BofA. Both Δ27BofA variants lack predicted TMS1, but have TMS2 and the C-terminal region proposed to

interact with SpoIVFA and SpoIVFB (*Figure 7C*), accounting for most of the steric hindrance mechanism of inhibition. BofA TMS1 appears to play a minor role by sterically hindering some interactions between the Pro-σ$^K$ N-terminal region and SpoIVFB, based on the comparison between full-length BofA and MBPΔ27BofA in cross-linking time courses. The interactions hindered more by full-length BofA included cross-links between the Pro-σ$^K$ Proregion and the SpoIVFB membrane-reentrant loop (*Figure 6G and H*), which may bind to the Proregion and present it to the SpoIVFB active site for cleavage (*Halder et al., 2017*) based on a study of the homologous membrane-reentrant loop of *E. coli* RseP (*Akiyama et al., 2015*). Full-length BofA also hindered cross-links between Pro-σ$^K$ L41C and the SpoIVFB interdomain linker (Y214C) more than MBPΔ27BofA (*Figure 6I*), suggesting that BofA TMS1 weakens the interaction between the interdomain linker and Pro-σ$^K$, which is crucial for cleavage (*Halder et al., 2017*; *Ramirez-Guadiana et al., 2018*). Our model predicts that both TMSs of BofA, as well as its C-terminal region, occupy the SpoIVFB active site cleft (*Figure 7B*), plausibly explaining the additional steric hindrance that BofA TMS1 appears to provide. The minor role of BofA TMS1 may be important since premature σ$^K$ production can reduce sporulation efficiency (*Cutting et al., 1990*).

## Implications for activation of SpoIVFB and its orthologs in other endospore formers

Activation of SpoIVFB during *B. subtilis* sporulation occurs when SpoIVB cleaves the C-terminal end of SpoIVFA (*Dong and Cutting, 2003*; *Campo and Rudner, 2006*; *Campo and Rudner, 2007*; *Mastny et al., 2013*) and CtpB cleaves the C-terminal ends of both SpoIVFA and BofA (*Campo and Rudner, 2006*; *Campo and Rudner, 2007*; *Zhou and Kroos, 2005*; *Figure 1—figure supplement 1*). The levels of inhibitory proteins decrease coincident with increasing Pro-σ$^K$ cleavage (*Kroos et al., 2002*; *Dong and Cutting, 2003*; *Campo and Rudner, 2006*; *Zhou and Kroos, 2005*), suggesting that degradation of SpoIVFA and BofA permits SpoIVFB activity. Alternatively, a conformational change in the inhibition complex may suffice to trigger SpoIVFB activity, based on in vitro experiments indicating that SpoIVB cleavage of SpoIVFA alters the susceptibility of the complex to general proteolysis but does not disrupt the complex (*Campo and Rudner, 2006*).

Our results imply that activation of SpoIVFB requires the removal or substantial movement of inhibitory proteins to allow substrate access to the active site. Our model predicts that SpoIVFA bridges the BofA C-terminal region and SpoIVFB TMS4 to stabilize the inhibition complex (*Figure 7C*). We propose that cleavage of SpoIVFA by SpoIVB initiates destabilization of the inhibition complex. The initial cleavage, perhaps followed by further proteolysis of SpoIVFA, may destabilize the membrane-embedded BofA C-terminal region and expose it to cleavage by CtpB, which hastens SpoIVFB activation but is not essential (*Pan et al., 2003*; *Campo and Rudner, 2006*; *Campo and Rudner, 2007*; *Zhou and Kroos, 2005*). Finally, degradation of BofA or a conformational change would relieve steric hindrance by BofA TMS2 and allow substrate access to the SpoIVFB active site.

Whether BofA and SpoIVFA completely prevent Pro-σ$^K$ from interacting with SpoIVFB prior to activation during sporulation is an open question. Pull-down and localization experiments suggested a complete block, but involved fusion proteins that may weaken the interaction between SpoIVFB and Pro-σ$^K$ (*Ramirez-Guadiana et al., 2018*). Our experiments employed cytTM-SpoIVFB variants and heterologous expression in growing *E. coli*. Pull-down and cross-linking results showed that inhibitory proteins do not prevent Pro-σ$^K$(1–127) from interacting with cytTM-SpoIVFB (*Figure 5* and *Figure 5— figure supplement 1*). Neither did GFPΔ27BofA and SpoIVFA prevent full-length Pro-σ$^K$ from inter-acting with cytTM-SpoIVFB (*Figure 5—figure supplement 2*), although the interaction appeared to be weakened by the presence of the C-terminal half of Pro-sigK. Perhaps full-length BofA and SpoIVFA would completely prevent full-length Pro-σ$^K$ from interacting with cytTM-SpoIVFB.

The mechanisms of inhibition and activation of *B. subtilis* SpoIVFB likely apply to its orthologs in spore-forming bacilli since these bacteria encode BofA, SpoIVFA, SpoIVB, and CtpB (*Galperin et al., 2012*; *Ramos-Silva et al., 2019*). This group includes well-known human or plant pathogens such as *B. anthracis*, *B. cereus*, and *B. thuringiensis*. In contrast, clostridia that form endospores do not have a recognizable gene for SpoIVFA (*de Hoon et al., 2010*; *Galperin et al., 2012*). In some clostridia, a nonorthologous gene has been proposed to code for a protein that performs the same function as SpoIVFA (*Galperin et al., 2012*), so similar mechanisms of SpoIVFB regulation may apply. The *E. coli* system described herein will facilitate further testing of heterologous protein function. Many spore-forming clostridia also lack a recognizable gene for BofA (*Ramos-Silva et al., 2019*). Our results imply that SpoIVFB activity is unregulated in these bacteria (i.e., coordination between FS and MC gene expression is lost). A precedent for less precise control of σ$^K$-dependent gene expression is known from studies of the human pathogen *Clostridioides difficile* (*Pereira et al., 2013*; *Saujet et al., 2013*), in which the gene for σ$^K$ does not encode a Proregion and a gene for SpoIVFB is absent (*Haraldsen and Sonenshein, 2003*). Based on phylogenomic analysis, the precise control of σ$^K$-dependent gene expression observed in *B. subtilis* emerged early in evolution and persisted in bacilli which propagate primarily in aerobic environments, but has been modified or lost in many clostridia which are found mainly in anaerobic habitats (*Ramos-Silva et al., 2019*).

## Design of IP Inhibitors

Efforts toward the development of therapeutic peptidyl inhibitors of serine IPs (rhomboids) are well-advanced (*Cho et al., 2019*; *Tichá et al., 2017*; *Tichá et al., 2018*). Translational work on aspartyl IPs has focused on γ-secretase owing to its processing of the amyloid precursor protein (APP) associated with Alzheimer's disease (*Wolfe, 2019*). Wide-spectrum inhibitors of γ-secretase exhibit toxicity in clinical trials, mainly due to inhibition of signaling via Notch receptors, which are also substrates of γ-secretase (*De Strooper and Chávez Gutiérrez, 2015*). The structures of γ-secretase complexes with APP and Notch reveal differences in binding that may allow substrate-specific inhibitors to be developed as therapeutics (*Yang et al., 2019*; *Zhou et al., 2019*). Stabilization of complexes in which γ-secretase progressively cleaves APP is another promising approach toward the development of drugs to treat Alzheimer's disease (*Szaruga et al., 2017*).

In comparison with rhomboids and γ-secretase, translational work on metallo IP inhibitors is in its infancy. Batimastat, a peptidic hydroxamate known to inhibit eukaryotic matrix metalloproteases (MMPs), selectively inhibited RseP in *E. coli* in a recent study (*Konovalova et al., 2018*). A lack of selectivity has impeded efforts toward using peptidic hydroxamates as MMP inhibitors for the treat-ment of cancer, arthritis, and other diseases (*Jacobsen et al., 2010*; *Fields, 2015*; *Gomis-Rüth, 2017*). The hydroxamate strongly coordinates the catalytic metal ion and the peptidic portions have lacked sufficient specificity to prevent off-target effects. Even so, side effects may not preclude using peptidic hydroxamates to treat bacterial infections topically or with brief systemic administration. Screening other small-molecule MMP inhibitors (*Jacobsen et al., 2010*) for activity against bacteria and fungi with metallo IPs known to be involved in pathogenesis (*Chang et al., 2007*; *Urban, 2009*; *Kroos and Akiyama, 2013*; *Schneider and Glickman, 2013*) may prove productive.

Our findings reveal concepts that may inform efforts to design selective inhibitors of IPs. BofA TMS2 appears to block the SpoIVFB active site (*Figure 7A*), similar to the prosegment of many latent proteases (*Arolas et al., 2018*). Like BofA, some prosegments can act in trans, an observation that led to prodomains being used as selective inhibitors of A Disintegrin And Metalloproteinase (ADAM) family enzymes (*Moss et al., 2007*; *Wong et al., 2016*). Selectivity relies on the extensive interaction

surface between the prodomain and the enzyme, including features specific to the pair, which can boost efficacy and diminish off-target effects in therapeutic strategies (*Gomis-Rüth, 2017*). A peptide corresponding to BofA N48 to T64 may inhibit SpoIVFB, but our model also predicts extensive interactions between the BofA C-terminal region (residues 65–87) and SpoIVFB (*Figure 7B*). A longer BofA peptide, perhaps flexibly linked to SpoIVFA residues 96–109, which are also predicted to interact with SpoIVFB (*Figure 7C*), may exhibit improved inhibition and selectivity. Testing peptide inhibitory activity in vitro is possible since purified cytTM-SpoIVFB cleaves Pro-$\sigma^K$(1-127) (*Zhou et al., 2009*). Structure determination of SpoIVFB in complex with an inhibitory peptide, GFPΔ27BofA, or full-length BofA is an important goal to facilitate the design of metallo IP inhibitors. In particular, it may inform efforts to design inhibitors of SpoIVFB orthologs in pathogenic bacilli and clostridia, which persist by forming endospores (*Al-Hinai et al., 2015*; *Checinska et al., 2015*; *Browne et al., 2016*). Such efforts could lead to new strategies to control endosporulation.

Our findings also suggest that BofA TMS1 interferes with interactions between the Pro-$\sigma^K$ N-terminal region and SpoIVFB. Therefore, a cyclic peptide inhibitor may prove to be more effective than a linear one. The desirable characteristics of cyclic peptides as therapeutics, and new methods of producing and screening cyclic peptide libraries, make this an attractive possibility (*Zorzi et al., 2017*; *Sohrabi et al., 2020*). Although we favor a simple steric hindrance mechanism of inhibition not involving a BofA residue acting as a zinc ligand, it is possible that inclusion of such a residue in a BofA TMS2 peptide mimic would improve SpoIVFB inhibition. A similar strategy could be used to design inhibitors of other metallo IPs.

# Materials and methods

**Key resources table**

| Reagent type (species) or resource | Designation | Source or reference | Identifiers | Additional information |
|---|---|---|---|---|
| Gene (*Bacillus subtilis*) | *bofA* | Subtiwiki | BSU_00230 | |
| Gene (*B. subtilis*) | *spoIIIC* (*sigK*) | Subtiwiki | BSU_26390 | 3' part of the interrupted sigma K gene |
| Gene (*B. subtilis*) | *spoIVCB* (*sigK*) | Subtiwiki | BSU_25760 | 5' part of the interrupted sigma K gene |
| Gene (*B. subtilis*) | *spoIVFA* | Subtiwiki | BSU_27980 | |
| Gene (*B. subtilis*) | *spoIVFB* | Subtiwiki | BSU_27970 | |
| Strain, strain background (*Escherichia coli*) | BL21(DE3) | Novagen | Cat# 69450 | Competent cells |
| Strain, strain background (*B. subtilis*) | PY79 | *Youngman et al., 1984* | | Prototrophic wild-type strain |
| Strain, strain background (*B. subtilis*) | BK754 | *Cutting et al., 1991a* | *spoIVB165* | |
| Strain, strain background (*B. subtilis*) | ZR264 | *Zhou and Kroos, 2004* | *spoIVB165 bofA::erm* | |
| Antibody | Anti-penta-His-HRP conjugate (Mouse monoclonal) | QIAGEN | Cat# 34460 | WB (1:10,000) |
| Antibody | Anti-FLAG M2-HRP conjugate (Mouse monoclonal) | Sigma-Aldrich | Cat# A8592 | WB (1:10,000) |
| Antibody | Anti-GFP (Rabbit polyclonal) | *Kroos et al., 2002* | | WB (1:10,000) |
| Antibody | Anti-MBP (Rabbit polyclonal) | NEB | Cat# E8030S | WB (1:10,000) |
| Antibody | Anti-SpoIVFA (Rabbit polyclonal) | *Kroos et al., 2002* | | WB (1:3000) |
| Antibody | Anti-SpoIVFB (Rabbit polyclonal) | *Yu and Kroos, 2000*; *Halder et al., 2017* | | WB (1:5000) |
| Antibody | Anti-Pro-$\sigma^K$ (Rabbit polyclonal) | *Lu et al., 1990* | | WB (1:3000) |
| Recombinant DNA reagent | pSO40 (plasmid) | This paper | pET Quartet | *Supplementary file 1* |

*Continued on next page*

*Continued*

| Reagent type (species) or resource | Designation | Source or reference | Identifiers | Additional information |
|---|---|---|---|---|
| Recombinant DNA reagent | pSO78 (plasmid) | This paper | P$_{bofA}$-GFPΔ27BofA | *Supplementary file 1* |
| Recombinant DNA reagent | pSO96 (plasmid) | This paper | Cys-less pET Duet | *Supplementary file 1* |
| Recombinant DNA reagent | pSO139 (plasmid) | This paper | Cys-less pET Quartet | *Supplementary file 1* |
| Commercial assay or kit | Quikchange | Stratagene | Cat# 200518 | |
| Chemical compound, drug | 1,10-phenanthroline monohydrate | Sigma-Aldrich | Cat# P9375-5G | Also called 2-phenanthroline |
| Chemical compound, drug | n-dodecyl-β-D-maltoside (DDM) | Anatrace | Cat# D310S | 1% |
| Software, algorithm | Image Lab | Bio-Rad | | Version 5.2.1 |
| Software, algorithm | ImageJ | (http://imagej.nih.gov/ij/) | | |
| Software, algorithm | T-Coffee | *Notredame et al., 2000*, (https://tcoffee.crg.eu) | | |
| Software, algorithm | UniClust30 database | (https://uniclust.mmseqs.com) | | Version 2020_02 |
| Software, algorithm | trRosetta | *Yang et al., 2020*, (https://github.com/gjoni/trRosetta) | | Model version 2019_07 |
| Software, algorithm | HHblits | *Remmert et al., 2011*, (https://github.com/soedinglab/hh-suite) | | Version 3.3.0 |
| Software, algorithm | PyRosetta | *Chaudhury et al., 2010*, (https://www.pyrosetta.org/) | | Version 2020.28 |
| Software, algorithm | CHARMM | (https://www.charmm.org) | | Version c42a2 |
| Other | MM 4–64 (FM 4–64) | AAT Bioquest | Cat# 21487 | (1 µg/ml) |
| Other | Anti-DYKDDDDK magnetic agarose | Pierce | Cat# A36797 | |
| Other | TALON Superflow Metal Affinity Resin | TaKaRa | Cat# 635507 | |

## Plasmids, primers, and strains

Plasmids used in this study are described in *Supplementary file 1*, as are primers used in plasmid construction. Plasmids were cloned in *E. coli* strain DH5α (*Hanahan, 1983*). Relevant parts of plasmids were verified by DNA sequencing with primers listed in *Supplementary file 1 B. subtilis* strains used in this study are also described therein.

## Pro-σ$^K$(1-127) cleavage in *E. coli*

Strain BL21(DE3) (Novagen) was used to produce proteins in *E. coli*. Two plasmids with different antibiotic resistance genes were cotransformed (*Zhou and Kroos, 2004*) or a single plasmid was transformed, with selection on Luria-Bertani (LB) agar supplemented with kanamycin sulfate (50 µg/ml) and/or ampicillin (100 µg/ml). Transformants (4–5 colonies) were grown in LB medium with 50 µg/ml kanamycin sulfate and/or 200 µg/ml ampicillin at 37°C with shaking (200 rpm). Typically, overnight culture (200 µl) was transferred to 10 ml of LB medium with antibiotics, cultures were grown at 37°C with shaking (250 rpm) to an optical density of 60–80 Klett units, and isopropyl β-D-thiogalactopyranoside (IPTG) (0.5 mM) was added to induce protein production for 2 hr. For transformants with either pET Quintet or full-length Pro-σ$^K$-His$_6$, overnight growth was avoided. Transformants were transferred directly to 10 ml of LB medium with antibiotic, and cultures were grown and induced as described above. Equivalent amounts of cells (based on optical density in Klett units) were collected (12,000×*g* for 1 min) and extracts were prepared (*Zhou and Kroos, 2004*), then subjected to immunoblot analysis.

## Immunoblot analysis

Samples were subjected to immunoblot analysis as described (*Kroos et al., 2002*). Briefly, proteins were separated by SDS-PAGE using Prosieve (Lonza) polyacrylamide gels (10% for disulfide cross-linking experiments and 14% for cleavage assays) and electroblotted to Immobilon-P membranes

(Millipore). Protein migration was monitored using SeeBlue Plus2 Prestained Standard (Invitrogen) and blots were blocked with 5% nonfat dry milk (Meijer) in TBST (20 mM Tris-HCl pH 7.5, 0.5 M NaCl, and 0.1% Tween 20) for 1 hr at 25°C with shaking. Blots were probed with antibodies against His$_6$ (penta-His QIAGEN catalog #34460; 1:10,000), FLAG$_2$ (Sigma-Aldrich catalog #A8592; 1:10,000), GFP (*Kroos et al., 2002*) (1:10,000), MBP (NEB catalog #E8030S; 1:10,000), SpoIVFA (*Kroos et al., 2002*) (1:3000), Pro-σ$^K$ (*Lu et al., 1990*) (1:3000), and/or SpoIVFB (*Yu and Kroos, 2000*; *Halder et al., 2017*) (1:5000) diluted in TBST with 2% milk, overnight at 4°C with shaking. Since the GFP, MBP, SpoIVFA, Pro-σ$^K$, and SpoIVFB antibodies were not HRP-conjugated, they were detected with goat anti-rabbit-HRP antibody (Bio-Rad catalog #170-6515; 1:10,000) diluted in TBST with 2% milk, 1 hr at 25°C with shaking. Signals were generated using the Western Lightning Plus ECL reagent (PerkinElmer) and detected using a ChemiDoc MP imaging system (Bio-Rad). Unsaturated signals were quantified using the Image Lab 5.2.1 software (Bio-Rad) lane and bands tool in order to determine the Pro-σ$^K$(1–127) cleavage ratio or the ratio of disulfide cross-linked complex to the total intensity of the cytTM-SpoIVFB variant monomer, dimer, and complex.

## BofA sequence analysis

Orthologs of *B. subtilis bofA*, which are present in the genomes of most endospore-forming bacteria (*Galperin et al., 2012*), were collected from the NCBI and Uniprot databases. The protein sequences of BofA orthologs were aligned using the T-Coffee multiple sequence alignment (MSA) package (*Notredame et al., 2000*). Residues identical in at least 70% of the sequences were considered conserved.

## *B. subtilis* sporulation and GFPΔBofA localization

GFPΔ27BofA or its variants were expressed under control of the *bofA* promoter (P$_{bofA}$) after chromosomal integration at the *amyE* locus. Plasmids bearing P$_{bofA}$-*gfpΔ27bofA* or its variant, bordered by regions of homology to *B. subtilis amyE*, were transformed into strain ZR264. Transformants with a gene replacement at *amyE* were selected on LB agar with spectinomycin sulfate (100 µg/ml) and identified by loss of amylase activity (*Harwood and Cutting, 1990*). Sporulation was induced by growing cells in the absence of antibiotics, followed by the resuspension of cells in SM medium (*Harwood and Cutting, 1990*). At indicated times PS, samples (50 µl) were centrifuged (12,000×*g* for 1 min), supernatants were removed, and cell pellets were stored at –80°C. Whole-cell extracts were prepared as described for *E. coli* (*Zhou and Kroos, 2004*), except samples were incubated at 50°C for 3 min instead of boiling for 3 min (*Halder et al., 2017*), and proteins were subjected to immunoblot analysis.

To image GFPΔ27BofA localization, samples collected at 3 hr PS were examined by fluorescence microscopy using an Olympus FluoView FV-1000 filter-based confocal microscope. GFPΔ27BofA (ex/em ~488/507 nm) was excited using a 458 nM argon laser and fluorescence was captured using a BA465–495 nm band pass filter. The lipophilic dye FM 4–64 (1 µg/ml) (AAT Bioquest) was used to stain membranes. FM 4–64 (ex/em ~515/640 nm) was excited using a 515 nm argon laser and fluorescence was captured using a BA560IF band pass filter (*Parrell and Kroos, 2020*).

## Disulfide cross-linking

A method described previously (*Koide et al., 2008*) was used with slight modifications (*Zhang et al., 2013*). As described above for Pro-σ$^K$(1–127) cleavage, *E. coli* BL21(DE3) was transformed with a plasmid, grown in LB (10 ml), induced with IPTG, and equivalent amounts of cells were collected. Cells were mixed with chloramphenicol (200 µg/ml) and 2-phenanthroline (3 mM), collected by centrifugation (12,000×*g* for 1 min), washed with 10 mM Tris-HCl pH 8.1 containing 3 mM 2-phenanthroline, and suspended in 10 mM Tris-HCl pH 8.1. Samples were treated with 1 mM Cu$^{2+}$(phenanthroline)$_3$ or 3 mM 2-phenanthroline (as a negative control) for 15, 30, 45, or 60 min at 37°C, followed by incubation with neocuproine (12.5 mM) for 5 min at 37°C. Cells were lysed and proteins were precipitated by the addition of trichloroacetic acid (5%) and inversion every 5 min for 30 min on ice. Proteins were sedimented by centrifugation (12,000×*g*) for 15 min at 4°C, the supernatant was removed, and the pellet was washed with cold acetone. The pellets were sedimented by centrifugation (12,000×*g*) for 5 min at 4°C and the supernatants were discarded. The pellets were dried for 5 min at 25°C and resuspended in buffer (100 mM Tris-HCl pH 7.5, 1.5% SDS, 5 mM EDTA, 25 mM *N*-ethylmaleimide) for 30 min at 25°C. Portions were mixed with an equal volume of sample buffer (25 mM Tris-HCl pH 6.8, 2% SDS, 10% glycerol, and 0.015% bromophenol blue) with or without 100 mM DTT, and were typically incubated

at 37°C for 10 min, prior to immunoblot analysis. In experiments with single-Cys V70C cytTM-SpoIVFB E44Q, single-Cys Pro-σ$^K$(1–127) variants, and Cys-less variants of BofA and SpoIVFA, samples were boiled 3 min prior to immunoblot analysis, which helped to resolve species (*Figure 6—figure supplement 3E and G*).

## Co-immunoprecipitation (FLAG$_2$ pull-down assays)

*E. coli* BL21(DE3) was transformed with a plasmid, grown in LB (1 L), and induced with IPTG as described above. The culture was split, cells were harvested, and cell pellets were stored at –80°C. Cell lysates were prepared as described (*Zhang et al., 2016*), except that each cell pellet was resuspended in 20 ml of lysis buffer containing 50 mM Tris-HCl pH 7.1 rather than PBS. Cell lysates were centrifuged (15,000×*g* for 15 min at 4°C) to sediment cell debris and protein inclusion bodies. The supernatant was treated with 1% *n*-dodecyl-β-D-maltoside (DDM) (Anatrace) for 1 hr at 4°C to solubilize membrane proteins, then centrifuged at 150,000×*g* for 1 hr at 4°C. The supernatant was designated the input sample and 1 ml was mixed with 50 µl anti-DYKDDDDK magnetic agarose (Pierce) that had been equilibrated with buffer (50 mM Tris-HCl pH 7.1, 0.1% DDM, 5 mM 2-mercaptoethanol, and 10% glycerol) and the mixture was rotated for 1 hr at 25°C. The magnetic agarose was removed with a DynaMag-2 magnet (Invitrogen) and the supernatant was saved (unbound sample). The magnetic agarose was washed three times by gently vortexing with 500 µl wash buffer (50 mM Tris-HCl pH 7.1, 150 mM NaCl, 10% glycerol, and 0.1% DDM), then washed once with 500 µl water. The magnetic agarose was mixed with 50 µl of 2× sample buffer (50 mM Tris·HCl pH 6.8, 4% SDS, 20% glycerol, 200 mM DTT 0.03% bromophenol blue) and boiled for 3 min (bound sample). A portion of the bound sample was diluted tenfold (1/10 bound sample) with 1× sample buffer to match the concentration of the input sample. Samples were subjected to immunoblot analysis.

## Cobalt affinity purification (His$_6$ pull-down assays)

Input sample (15 ml) prepared as described above was mixed with imidazole (5 mM) and 0.5 ml of Talon superflow metal affinity resin (Clontech) that had been equilibrated with buffer (as above for magnetic agarose). The mixture was rotated for 1 hr at 4°C. The cobalt resin was sedimented by centrifugation at 708×*g* for 2 min at 4°C and the supernatant was saved (unbound sample). The resin was washed three times with 5 ml wash buffer (as above plus 5 mM imidazole), each time rotating the mixture for 10 min at 4°C and sedimenting resin as above. The resin was mixed with 0.5 ml 2× sample buffer and boiled for 3 min (bound sample). A portion of the bound sample was diluted 15-fold (1/15 bound sample) with 1× sample buffer to match the concentration of the input sample. Samples were subjected to immunoblot analysis.

## Modeling of complexes containing SpoIVFB, BofA, and parts of SpoIVFA and Pro-σ$^K$

The modeling proceeded through stages where initial monomeric models were assembled step-by-step into multimeric complexes guided primarily by the restraints from cross-linking experiments and predicted contacts from co-evolutionary coupling analysis. More specifically, a SpoIVFB monomer was first assembled from the membrane and CBS domains. Two monomers were combined into a dimer and the dimer was assembled into a plausible tetramer. Part of one molecule of Pro-σ$^K$ (residues 1–114) was subsequently added, which resulted in 'fb.sigk.' The 'fb.sigk.bofa.fa' model was developed by starting from 'fb.sigk,' truncating the Pro-σ$^K$ N-terminus (residues 1–37), and subsequently adding first BofA and finally SpoIVFA (residues 65–111).

   The membrane domain of a SpoIVFB monomer was modeled initially based on the structure for the site-2 protease from *Methanocaldococcus jannaschii* (PDB code: 3B4R) (*Feng et al., 2007*). The structure was combined with a CBS domain modeled based on the CBS-domain protein TM0935 from *Thermotoga martima* (PDB code: 1O50) (*Miller et al., 2004*). The CBS domain provides a dimerization interface for the C-terminal part of SpoIVFB. The full-length dimer, including the membrane domain, was completed by considering predicted contacts from co-evolutionary coupling analysis. A tetramer was then built guided by the arrangement of transmembrane helices and orientation of CBS domains in the recently published full-length structure of the chloride proton exchanger CLC-7 (PDB code: 7JM6) (*Schrecker et al., 2020*) and again by considering contacts from co-evolutionary couplings. We note that the resulting dimer and tetramer models are different from our previous model for the

SpoIVFB as we could take advantage of the new structural template and additional information from the co-evolutionary couplings. The initial model for the well-folded C-terminal part of Pro-σ$^K$ (residues 40–114) was built based on the structure of RNA polymerase sigma subunit domain 2 (PDB code: 3UGO) (*Feklistov and Darst, 2011*). Suitable templates are not available for BofA and the part of SpoIVFA modeled here. For these components, initial models were obtained based on the predicted intra-chain contacts from the co-evolutionary couplings.

The modeling under restraints was initially carried out using Cα-based coarse-grained models that were allowed to relax under restraints via cycles of energy minimization and short molecular dynamics simulations at elevated temperature as described in more detail previously (*Halder et al., 2017*). Restraints from cross-linking were applied as described previously (*Halder et al., 2017*). Different from our previous work, we also applied extensive intra- and inter-chain restraints based on predicted contacts from co-evolutionary coupling analysis. In addition, we applied positional restraints to keep the structures of the SpoIVFB membrane and CBS domains and the folded C-terminus of Pro-σ$^K$ close to their initial models while allowing subunits to move relative to each other as guided by the restraints. For BofA and SpoIVFA, individual helices were restrained internally but allowed to move relative to each other to find the optimal arrangement in complex with SpoIVFB, again guided by the predicted contact restraints. Contact restraints with respect to SpoIVFB were implemented as minimum distance restraints to the closest residue in any of the four subunits since contact predictions cannot distinguish between chemically equivalent oligomer units. All available experimental cross-links were applied but the list of predicted contacts was edited to exclude certain contacts incompatible with cross-linking data and previous biochemical data on the overall topology of the SpoIVFB-BofA-SpoIVFA complex when embedded into the membrane. Excluded contacts may reflect uncertainties in the prediction method as inter-chain contacts are more difficult to predict reliably. They may also indicate alternate biologically relevant interactions between SpoIVFB, BofA, and SpoIVFA that were not probed in the experiments via cross-linking. Detailed lists of all applied restraints are given in *Supplementary files 2-5*.

Inter-residue contacts were predicted by trRosetta (*Yang et al., 2020*). A MSA, the input of trRosetta, for a protein was generated using HHblits (*Remmert et al., 2011*) to search against the UniClust30 database (*Mirdita et al., 2017*). It was further curated to only include sequences that are related to spore formation and are from organisms that have both SpoIVFB and SpoIVFA sequences. To predict inter-protein contacts using trRosetta, a hybrid MSA was generated by pasting two interacting proteins' sequences that are from the same strain. Twenty glycines were inserted between the target sequences to prevent irrelevant predictions near the pasted regions due to the proximity in the hybrid sequence. Similarly, gaps were inserted for the homologous sequences to preserve alignment with the *B. subtilis* proteins. The contact predictions for the inserted residues were ignored for further modeling.

Finally, the Cα models for the complexes were converted to an all-atom model using PyRosetta (*Chaudhury et al., 2010*) with the curated predicted intra- and inter-protein contacts and the cross-linking constraints. First, the model was locally minimized in the Rosetta centroid representation. The cross-linking constraints and inter-protein contacts were used for the Cα model building. Predicted intra-protein contacts were also used only if their contact probabilities were higher than 0.15 and if they were not violated severely in the Cα model; a predicted contact was considered as a severe violation if its score was greater than 10 Rosetta energy units (REUs). The relative weights for each scoring term were 25, 0.25, and 0.1 for the cross-linking constraints, inter-protein contacts, and intra-protein contacts, respectively. To prevent large deviation from the Cα model, harmonic restraints were applied on every Cα atom with a force constant of 0.1 REU/Å$^2$. In addition, Rosetta centroid energy terms were also used: Ramachandran energy (1.0), omega angle potential (0.5), backbone hydrogen bond energy (5.0), and van der Waals energy (1.0) with weights in the parentheses. The minimization was performed for 100 steps. Then, the minimized structure was converted to an all-atom model, and the FastRelax protocol was applied to the model with the Rosetta scoring function (ref2015) (*Park et al., 2016*) and the same cross-linking constraints, the contact predictions, and the harmonic restraints on Cα atoms that were used for the minimization. Eight all-atom models were generated from the Cα model, and a model with the least cross-linking constraint violations was selected. To generate final models, Zn$^{2+}$ ions were added at the active sites of each subunit of SpoIVFB to be coordinated with D137, H43, and H47 followed by another brief

minimization using CHARMM under harmonic restraints on heavy atoms to accommodate the $Zn^{2+}$ ions without clashes.

## Acknowledgements

The authors thank Daniel Parrell for assistance with fluorescence microscopy, Jon Kaguni for helpful comments on the manuscript, and David Rudner, Simon Cutting, Ruanbao Zhou, and Yang Zhang for plasmids. This study was supported by National Institutes of Health Grants R01 GM43585 (to LK) and R35 GM126948 (to MF), and by Michigan State University AgBioResearch.

## Additional information

### Funding

| Funder | Grant reference number | Author |
| --- | --- | --- |
| National Institutes of Health | R01 GM43585 | Sandra Olenic<br>Lee Kroos |
| National Institutes of Health | R35 GM12648 | Lim Heo<br>Michael Feig |

The funders had no role in study design, data collection and interpretation, or the decision to submit the work for publication.

### Author contributions

Sandra Olenic, Conceptualization, Data curation, Formal analysis, Investigation, Methodology, Validation, Visualization, Writing - original draft, Writing - review and editing; Lim Heo, Conceptualization, Data curation, Formal analysis, Investigation, Methodology, Visualization, Writing - original draft; Michael Feig, Conceptualization, Data curation, Formal analysis, Funding acquisition, Investigation, Methodology, Supervision, Visualization, Writing - original draft, Writing - review and editing; Lee Kroos, Conceptualization, Formal analysis, Funding acquisition, Project administration, Supervision, Visualization, Writing - review and editing

### Author ORCIDs

Sandra Olenic (ORCID) http://orcid.org/0000-0001-8578-9167
Lee Kroos (ORCID) http://orcid.org/0000-0002-4294-948X

### Decision letter and Author response

Decision letter https://doi.org/10.7554/eLife.74275.sa1
Author response https://doi.org/10.7554/eLife.74275.sa2

## Additional files

### Supplementary files

- Supplementary file 1. Plasmids, primers, and *B. subtilis* strains used in this study.

- Supplementary file 2. Applied restraints from cross-linking, and distance evaluation, for the model of SpoIVFB in complex with part of Pro-$\sigma^K$.

- Supplementary file 3. Applied restraints from co-evolutionary coupling analysis, and distance evaluation, for the model of SpoIVFB in complex with part of Pro-$\sigma^K$.

- Supplementary file 4. Applied restraints from cross-linking, and distance evaluation, for the model of SpoIVFB in complex with BofA and parts of SpoIVFA and Pro-$\sigma^K$.

- Supplementary file 5. Applied restraints from co-evolutionary coupling analysis, and distance evaluation, for the model of SpoIVFB in complex with BofA and parts of SpoIVFA and Pro-$\sigma^K$.

- Transparent reporting form

### Data availability

All data generated or analysed during this study are included in the manuscript and supporting files.

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
