## [Editor Report]

A member of a family of metalloproteases conserved in all three domains of life, SpoIVFB is required for development in the spore-forming firmicute, *Bacillus subtilis*. SpoIVFB activity is tightly controlled, however, its status as a multipass membrane protein has made illuminating the molecular basis of SpoIVFB inhibition challenging. Here, Olenic and colleagues combine genetics, cross-linking, and co-evolutionary analysis to develop a structural model of interaction between SpoIVFB and its inhibitors SpoIVFA and BofA. Given the conservation and importance of this family of metalloproteases, this work should have a broad impact influencing our understanding of the regulation of this class of proteins across the tree of life.

---

## [Decision Letter]

**Decision letter after peer review:**

Thank you for submitting your article "Inhibitory proteins block substrate access by occupying the active site cleft of *Bacillus subtilis* intramembrane protease SpoIVFB" for consideration by *eLife*. Your article has been reviewed by 3 peer reviewers, including Petra Levin as the Reviewing Editor and Reviewer #1, and the evaluation has been overseen by Volker Dötsch as the Senior Editor. The following individual involved in review of your submission have agreed to reveal their identity: Volker Dötsch (Reviewer #3).

Essential revisions:

All three reviewers appreciated the rigorous and thorough nature of the experiments. The only additional experiment requested is to assess protein levels for the mutant variants to determine if they accumulate to wild type concentrations. Additional suggestions to improve clarity and presentation are detailed in the reviews below.

*Reviewer #1 (Recommendations for the authors):*

As written the manuscript is dense and challenging to absorb and will need to be revised for a broad audience prior to publication. Below are some comments directed at condensing and clarifying different aspects of the manuscript.

Specific comments aimed at making the manuscript more accessible:

1. Generally, please change all passive voice to active voice in main text (it is fine in the Materials and methods section). Stating things in first person will eliminate a significant amount of the extra verbiage and streamline the manuscript.

2. Similarly, make sure topic sentences summarize the focus of each paragraph. The rationale for all experiments should be clearly stated.

3. Line 17-N-terminal pro region needs to be defined in abstract.

4. Line 76--This paragraph is a very long and detailed list of all the results. Please condense to important/key take home messages. There is no reason to include these details here as they are in the following Results section.

5. Line 100-Please write out acronyms whenever possible. It took a while to figure out what TMS was. Transmembrane sequence doesn't take that much additional space and does not require reader to remember.

6. Line 111. There is a lot of detail provided about a system that didn't end being used. Fine to say that authors developed the pET quartet plasmids to overcome technical issues associated with plasmid copy number. (put the details of construction in methods)

7. Line 129-what is meant by partial relief? Please provide fraction/percent relief in text.

8. Line 141-please be specific about why the lack of conserved residues allows GFP∆27 to be functional. Unclear from text.

9. Line 147 "The Ala substitutions were made in GFPΔ27BofA and co-produced with Pro-σK(1-127), cytTM-SpoIVFB, and SpoIVFA from pET Quartet plasmids in *E. coli*." seems to belong in previous paragraph. It is not a topic sentence.

10. Line 166-please clarify what is meant by "ectopically"? From a non-native locus? At the wrong stage of development?

11. Line 228 I believe is the topic sentence for the next paragraph? "Based on these data we hypothesized that BofA48 is a zinc ligand and…"

12. Line 240 Please provide rationale for this experiment. Consider using previous sentence (at end of previous paragraph) as start of this paragraph to clarify thinking here.

*Reviewer #2 (Recommendations for the authors):*

Much of the work has been performed carefully and conclusions are warranted. However, the instability of the proteins in the absence of the complex makes some of these studies difficult to interpret. The authors should consider comparing expression levels of BofA mutants alone in the absence of other complex proteins. This would help clarify if changes in BofA levels are due to effects of mutations on protein stability or complex stability.

It is unclear to me in the crosslinking studies why the P135C-SpoIVFG-BofA complex is not DTT sensitive.

*Reviewer #3 (Recommendations for the authors):*

1) The authors use alanine scanning to identify N48 as a critical amino acid for the inhibitory process. They first speculate that N48 complexes the essential zinc ion and replaces the necessary water molecule. This would indeed be a very interesting mechanism. They later however show based on their modelling approach that N48 more likely makes intra-protein contacts to stabilize the conformation of BofA. This is quite confusing and I recommend to remove the speculation about the zinc binding mechanism from the results part and only describe the result that N48 is found to be important. Speculation about the mechanism could then be placed in the discussion.

2) Related: it would be nice to have a figure that shows how the side chains of the three important residues interact. At the moment only backbone conformations are shown.

3) In some of the figures showing the results of the cross linking experiments the largest density is seen as diffuse bands of molecular weights higher than the complex. (e.g. Figure 4—figure supplement 4). Any explanation? Could this be non specific cross linking effects?

---

## [Author Response]

Essential revisions:All three reviewers appreciated the rigorous and thorough nature of the experiments. The only additional experiment requested is to assess protein levels for the mutant variants to determine if they accumulate to wild type concentrations. Additional suggestions to improve clarity and presentation are detailed in the reviews below.

We thank the reviewers for their appreciation of the experiments.

To address the request for an additional experiment, we created plasmids to express the N48A, N61A, and T64A variants of GFPΔ27BofA in *E. coli* in the absence of other *B. subtilis* proteins. Accumulation of the variants was indistinguishable from wild-type GFPΔ27BofA (new Figure 2—figure supplement 2). Therefore, the altered protein concentrations observed for the N61A and T64A variants and for SpoIVFA and cytTM-SpoIVFB in combination with all three variants depended on co-production with the other *B. subtilis* proteins (Figure 2), perhaps indicative of altered complex formation leading to protein instability (l. 155-159). The new result suggests that the GFPΔ27BofA variants fold almost normally, but cause subtle changes in a BofA structural domain including TMS2 and a membrane-embedded C-terminal region, altering formation of the SpoIVFB inhibition complex (l. 473-487).

We thank the reviewers for their suggestions to improve clarity and presentation. We describe our responses below.

Reviewer #1 (Recommendations for the authors):As written the manuscript is dense and challenging to absorb and will need to be revised for a broad audience prior to publication. Below are some comments directed at condensing and clarifying different aspects of the manuscript.Specific comments aimed at making the manuscript more accessible:1. Generally, please change all passive voice to active voice in main text (it is fine in the Materials and methods section). Stating things in first person will eliminate a significant amount of the extra verbiage and streamline the manuscript.

We changed nearly all passive voice to active voice in the main text.

2. Similarly, make sure topic sentences summarize the focus of each paragraph. The rationale for all experiments should be clearly stated.

We made sure topic sentences summarize the focus of each paragraph (e.g., l. 99, 106, and elsewhere).

3. Line 17-N-terminal pro region needs to be defined in abstract.

Rather than defining Proregion in the Abstract, we deleted it there and defined it as the “pro-sequence region (Proregion)” in the Introduction (l. 75).

4. Line 76--This paragraph is a very long and detailed list of all the results. Please condense to important/key take home messages. There is no reason to include these details here as they are in the following Results section.

We deleted the first 10 lines of the paragraph, added a short topic sentence, and left the important/key take home messages.

5. Line 100-Please write out acronyms whenever possible. It took a while to figure out what TMS was. Transmembrane sequence doesn't take that much additional space and does not require reader to remember.

TMS is defined in the second sentence of the Introduction (l. 29) and used four more times in the Introduction, and 84 times in total, so we did not write it out each time. However, since TMS1 and TMS2 are used frequently, we did write out each at first occurrence (l. 74 and 95).

6. Line 111. There is a lot of detail provided about a system that didn't end being used. Fine to say that authors developed the pET quartet plasmids to overcome technical issues associated with plasmid copy number. (put the details of construction in methods)

We deleted the details (l. 99-101). We describe plasmid construction in Supplementary File 1 and induction of protein production in *E. coli* in Materials and methods (l. 617-625).

7. Line 129-what is meant by partial relief? Please provide fraction/percent relief in text.

We deleted “partial relief” from the revised paragraph topic sentence (l. 122-123). We define “partial relief” and the percent by stating “partially relieved cytTM-SpoIVFB inhibition, resulting in 14% and 5% more cleavage…” (l. 125-128) and “partially relieved inhibition, resulting in 12% more cleavage…” (l. 130-131).

8. Line 141-please be specific about why the lack of conserved residues allows GFP∆27 to be functional. Unclear from text.

We revised the sentence and added a sentence to clarify the observations that suggest BofA “TMS1 plays a minor role in SpoIVFB inhibition and that any residues compatible with TMS formation may suffice for that role.” (l. 140-144).

9. Line 147 "The Ala substitutions were made in GFPΔ27BofA and co-produced with Pro-σK(1-127), cytTM-SpoIVFB, and SpoIVFA from pET Quartet plasmids in *E. coli*." seems to belong in previous paragraph. It is not a topic sentence.

We agree. We moved the sentence to the end of the previous paragraph (l. 149-151).

10. Line 166-please clarify what is meant by "ectopically"? From a non-native locus? At the wrong stage of development?

We removed “ectopically” and clarified that we mean “non-native chromosomal locus” (l. 172) and “from the non-native *amyE* chromosomal locus” (l. 177). We also revised the paragraph to better explain the experiment.

11. Line 228 I believe is the topic sentence for the next paragraph? "Based on these data we hypothesized that BofA48 is a zinc ligand and…"

We removed the hypothesis that BofA N48 is a zinc ligand from the Results, at the suggestion of Reviewer #3.

12. Line 240 Please provide rationale for this experiment. Consider using previous sentence (at end of previous paragraph) as start of this paragraph to clarify thinking here.

As mentioned in our preceding response, we removed the hypothesis that BofA N48 is a zinc ligand. We broadened the hypothesis – “TMS2 of BofA occupies the SpoIVFB active site cleft in the inhibition complex” and explained the rationale for the hypothesis – “the GFPΔ27BofA N48 side chain is important for inhibition (Figure 2 and 3) and located near the middle of predicted TMS2 (Figure 2—figure supplement 1).” (l. 225-228). We then explain in general terms a strategy to begin testing the hypothesis based on a model of the SpoIVFB active site cleft and reasoning that BofA TMS2 occupancy of the cleft may be detectable using a disulfide cross-linking approach (l. 228-236). Because this approach may be unfamiliar to readers and we use it in many subsequent experiments, we added a sentence explaining its implementation in the context of the SpoIVFB inhibition complex (l. 236-239). Finally, we describe the first experiment to begin testing the hypothesis (l. 240-249) and the result (250-261).

Reviewer #2 (Recommendations for the authors):Much of the work has been performed carefully and conclusions are warranted. However, the instability of the proteins in the absence of the complex makes some of these studies difficult to interpret. The authors should consider comparing expression levels of BofA mutants alone in the absence of other complex proteins. This would help clarify if changes in BofA levels are due to effects of mutations on protein stability or complex stability.

We think the reviewer meant “the potential instability of the proteins in the absence of the complex…”. As mentioned in our response to “Essential Revisions”, we expressed the N48A, N61A, and T64A variants of GFPΔ27BofA in *E. coli* in the absence of other *B. subtilis* proteins and found that accumulation of the variants was indistinguishable from wild-type GFPΔ27BofA (new Figure 2—figure supplement 2). Since the Ala substitutions had little or no effect on GFPΔ27BofA stability, altered complex formation leading to protein instability likely explains the altered protein concentrations observed upon co-production with the other *B. subtilis* proteins (Figure 2) (l. 155-159).

During sporulation of *B. subtilis*, the three variants of GFPΔ27BofA accumulated normally at 4 h poststarvation, yet allowed premature cleavage of Pro-σ^K^, despite reducing the level of SpoIVFB and in some cases SpoIVFA (Figure 3A). These observations support altered complex formation leading to instability of SpoIVFB and in some cases SpoIVFA, without additional experiments.

Together, the data suggest that the GFPΔ27BofA variants fold almost normally in *E. coli* in the absence of other *B. subtilis* proteins (new Figure 2—figure supplement 2) and at 4 h during *B. subtilis* sporulation (Figure 3A), but subtle changes alter formation of the inhibition complex, relieving SpoIVFB inhibition and reducing its level. In the Discussion (l. 473-516), we integrate these observations with others that can be explained by proposing that “the BofA N48, N61, and T64 side chains are critical for interactions that stabilize a structural domain including TMS2 and a membrane-embedded C-terminal region, which in turn interact with SpoIVFA and SpoIVFB (Figure 7C), accounting for most of the steric hindrance mechanism of inhibition.” (l. 473-476).

It is unclear to me in the crosslinking studies why the P135C-SpoIVFG-BofA complex is not DTT sensitive.

We think the reviewer is referring to the bands labeled “complex” in lanes 8 and 9 of Figure 4B. We find that cross-link reversal by DTT treatment is variable. For example, in Figure 6—figure supplement 4D (60 min), we observed slightly more reversal of the same cross-link by DTT treatment. In Figure 6—figure supplement 4A (60 min), we observed less reversal of the E44C cytTM-SpoIVFB cross-link to MBPΔ27BofA C46 by DTT treatment than in lanes 1 and 2 of Figure 4A. We observed many other examples in our cross-linking experiments, but we do not understand the variability. We decided not to comment on the variability in the manuscript, since we think it would distract readers. In general, some cross-links appear to be less sensitive to DTT treatment, perhaps owing to the chemical environment of the cross-link.

Reviewer #3 (Recommendations for the authors):1) The authors use alanine scanning to identify N48 as a critical amino acid for the inhibitory process. They first speculate that N48 complexes the essential zinc ion and replaces the necessary water molecule. This would indeed be a very interesting mechanism. They later however show based on their modelling approach that N48 more likely makes intra-protein contacts to stabilize the conformation of BofA. This is quite confusing and I recommend to remove the speculation about the zinc binding mechanism from the results part and only describe the result that N48 is found to be important. Speculation about the mechanism could then be placed in the discussion.

As mentioned in our response to comment 12 of Reviewer #1, we removed the hypothesis that BofA N48 is a zinc ligand from the Results. There, we broadened the hypothesis – “TMS2 of BofA occupies the SpoIVFB active site cleft in the inhibition complex” and explained the rationale for the hypothesis – “the GFPΔ27BofA N48 side chain is important for inhibition (Figure 2 and 3) and located near the middle of predicted TMS2 (Figure 2—figure supplement 1).” (l. 225-228).

In the Discussion, we note that “In addition to steric hindrance, inhibition of soluble metalloproteases often involves a residue of the inhibitory protein, prodomain, or prosegment with a side chain that coordinates the catalytic metal ion in place of a water molecule necessary for substrate peptide bond hydrolysis 6970(, ).” (l. 459-462), then we explain why neither BofA C46 nor N48 likely acts as a zinc ligand (l. 463-472).

2) Related: it would be nice to have a figure that shows how the side chains of the three important residues interact. At the moment only backbone conformations are shown.

We added a figure with images of the BofA model showing the N48, N61, and T64 side chains (Figure 7—figure supplement 2). We note that in the model, N48, N61, and T64 are in proximity and “likely able to interact, but the constraints from experiments and co-evolutionary couplings are not sufficient for predicting the exact side chain orientations with high certainty. So, even though the N61 side chain is shown pointing away from the N48 and T64 side chains in the model, slight structural rearrangements within the modeling constraints could allow the N61 side chain to interact more directly with the N48 and/or T64 side chain(s).” (l. 410-414).

As mentioned in our response to the first comment of Reviewer #2, we propose in the Discussion that “the BofA N48, N61, and T64 side chains are critical for interactions that stabilize a structural domain including TMS2 and a membrane-embedded C-terminal region, which in turn interact with SpoIVFA and SpoIVFB (Figure 7C), accounting for most of the steric hindrance mechanism of inhibition.” (l. 473-476).

3) In some of the figures showing the results of the cross linking experiments the largest density is seen as diffuse bands of molecular weights higher than the complex. (e.g. Figure 4—figure supplement 4). Any explanation? Could this be non specific cross linking effects?

We think the reviewer is referring to diffuse bands migrating between 45 and 55 kDa primarily in the lanes treated with Cu but not DTT in Figure 4—figure supplement 4. We agree that nonspecific cross-linking of the single-Cys cytTM-SpoIVFB variants to *E. coli* proteins is the most likely explanation, so we annotated the figure and added “The star (*) indicates likely nonspecific cross-linking of the single-Cys cytTM-SpoIVFB variants to *E. coli* proteins.” in the legend. We note that the four novel species starred in Figure 6—figure supplement 3 E and G likely have a similar explanation (l. 361-363).